# The dynamics of peak head responses at Dutch canal dikes and the impact of climate change

Bart Strijker[1,2], Matthijs Kok[1,2]

[1]Hydraulic Engineering, Delft University of Technology, Stevinweg 1, 2628 CN Delft, The Netherlands
[2]Risk and disaster management Unit, HKV Consultants, Lelystad, the Netherlands

*Correspondence to*: Bart Strijker (b.strijker@tudelft.nl)

**Abstract.** Managing the water and flood risk in low-lying polder regions depends on the performance of canal dikes. This performance is influenced by hydraulic heads, which can peak due to heavy rainfall, affecting their stability and potentially inducing dike breaches. Variations in head responses and head statistics are relevant for regional flood risk analysis of canal 10   dike systems. This study examined the dynamics of peak heads in canal dikes on a national scale using time series models calibrated on a unique dataset on head observations across the dike system. Various model structures were evaluated, and a nonlinear model performed the best. These models were used to simulate 30 years of head time series representing current and future climate scenarios. Subsequently, dike clusters were identified based on the coincidence of peak heads, allowing for the identification of dikes where peaks are caused by (dis)similar types of rainfall events. The differences and similarities 15   in peak head response between dikes and identified clusters were related to physical dike characteristics. While the subsurface material and dike width appeared to influence the head response variation of clusters, their presence across multiple clusters indicates they do not yield a definitive outcome. Moreover, peak head statistics across various dikes indicated that extreme and yearly occurring load conditions are relatively close to each other, with a median decimate height of only 15 cm. With climate change driving higher winter precipitation and summer evaporation, head statistics are 20   changing. By 2100, extreme peak heads are expected to occur between 3 times less and 8 times more frequently, depending on the climate scenario and the type of canal dike.

## 1. Introduction

Catastrophic dike failures have occurred throughout history due to various causes, such as storm surges, extreme river discharges, ice drifts, and extreme weather conditions like heavy rainfall or drought. Several failure mechanisms were involved, including overflow and overtopping, external erosion, piping, and inner slope instability (Van Baars & Van Kempen, 2009; Özer et al., 2019b). For many dikes along rivers and coasts, inner slope instability occurs due to the infiltration of water into the dike body and its foundation, leading to higher head levels and pore-water pressures, reducing

effective stresses and shear strength of the soil (Frank et al., 2004; Ridley et al, 2004; Sharp et al., 2013; van Woerkom et al., 2021; Van der Krogt et al., 2022). The infiltration of water can be caused by high water levels against the dike as well as heavy rainfall (Rikkert, 2022; Van Baars & Van Kempen, 2009). For dikes with controlled water levels that show little fluctuations, such as canal dikes, the infiltration of water caused by heavy rainfall can be significant and is considered a primary mechanism of dike failure. Canal dikes are among others present in polders, which can be found in coastal and

alluvial lowlands all over the world, like the Netherlands, Bangladesh, Vietnam, England and China (Martín-Antón et al.,2016; Morton & Olson, 2018; Lendering et al., 2018; Tran & Weger, 2018; Triet et al., 2018; Manh et al., 2013; Warner et al., 2018). The water levels in these reclaimed areas are artificially regulated by an internal drainage system with canals, where water levels can reach several meters above the surrounding terrain (see Fig. 1). This makes these low-lying areas vulnerable to floods in the event of a canal dike breach. These canal dikes are found not only in polders but also along

internal waterways and irrigation canals worldwide, where major dike failures have occurred throughout history (Gildeh et al., 2019).

To manage flood risks in an embanked area, the performance of dikes plays a crucial role. The failure probabilities of the individual elements or dike stretches contribute to the reliability of the canal dike system and the flood risk level in an area

(Vanmarcke, 1977; Kanning, 2012; Jongejan et al., 2020). The Netherlands has an extensive system of canal dikes critical for managing its low-lying land and protecting it against flooding, with a total length of more than 10,000 km (Pleijster and van der Veeken, 2015). In general, the failure probability of a dike system increases with the total length of the dike system, due to partial correlation or independency between different individual dike stretches (Kanning, 2012; Vrijling & van Gelder, 2002). This phenomenon is known as the length effect, which is caused by both spatial dependencies in the

resistance and loading conditions and differs for each failure mechanism. Inner slope instability is a failure mechanism of the Dutch canal dikes that contributes significantly to the calculated failure probability and flood risks in the polders, where the load primarily consists of high hydraulic head peaks (Lendering et al., 2018; Rikkert et al., 2022; Van Baars and Van Kempen, 2009). The variations of the canal water levels in the drainage systems of Dutch polders are small (typically up to tens of centimeters), while the observed hydraulic head fluctuations are an order of magnitude larger than water level

fluctuations. Therefore, the fluctuations of the hydraulic heads are primarily driven by rainfall and evaporation. Whether two nearby canal dikes both experience extreme load conditions after a heavy rainfall event depends on the head response of the

dikes. Variations in the head response can cause extreme loads to occur after different weather events and influence the system's reliability. Furthermore, these variations in response also help water authorities identify threatening situations, as weather forecasts can be translated to potential peak head levels in dikes.


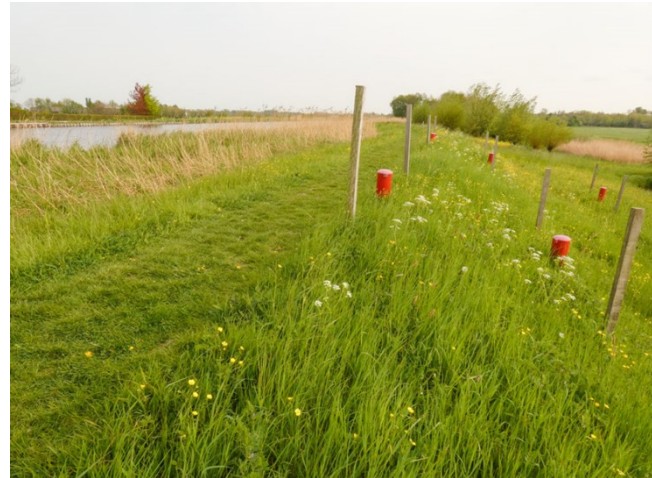 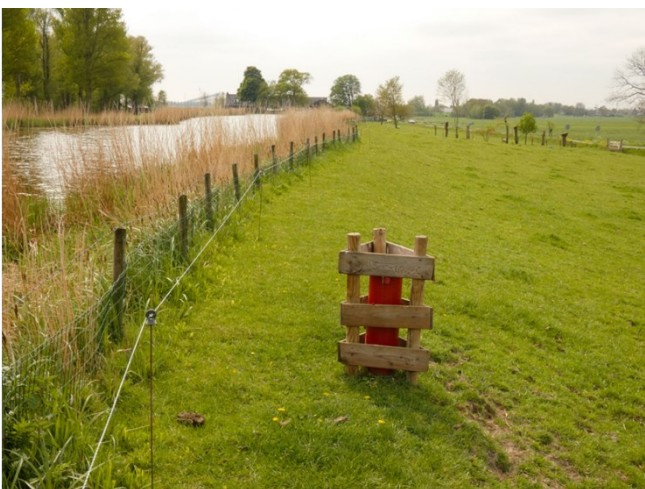

**Figure 1 Two examples of canal dikes in the Netherlands: Duifpolder is shown at the left and the Drooggemaakte Geer- en Kleine Blankaardpolder is depicted at the right. In both images, the canal has permanent high water levels close to the crest level (on the left side) and the low-lying polder is mainly used for agriculture (on the right side). The head differences between the canal water levels and water levels in the polders are 2.7m (Duifpolder) and 4.3m (Drooggemaakte Geer- en Kleine Blankaardpolder). The red tubes protect the measuring equipment and the piezometers used to measure the hydraulic head. Photos by EURECO/Cyril Liebrand, 2022.**


To calculate the failure probability of individual canal dikes and dike systems, information about peak head responses is essential. Currently, there is limited understanding regarding the spatial variability in head responses and head statistics in

canal dikes, which is partly due to the lack of measurements and the extensive computation time required for groundwater models. Multiple studies have modelled the effects of rainfall and evaporation on the phreatic surface in dikes using different approaches (Rikkert, 2022; Jamalinia et al., 2019; van Esch, 2012). Multi-year measurements of hydraulic head levels are often lacking in dikes, which makes modelling exercises difficult to validate. Furthermore, the validation of models is hindered by the heterogeneity of dikes and the unknown field hydraulic conductivities, potentially influenced by burrowing

animals, plant roots or cracks, and resulting flow paths. This makes that there is also little known about the effects of climate change on head statistics and failure probabilities of canal dikes. Future climate projections indicate increasing temperatures with summers becoming hotter and drier, and winters becoming warmer and wetter. This is expected to affect the stability of slopes. Although the impact has been studied for both natural slopes (e.g. Moore et al. 2010) and earthworks (e.g. Huang et al., 2024; Rouainia et al. 2020), for canal dikes, with different boundary conditions, subsurface materials and resulting head

dynamics, the effects are studied to a limited extent. This study aims to assess the dynamics of peak hydraulic heads in canal dikes on a national level, caused by heavy rainfall events, by analysing the variation in head responses and head statistics. It also seeks to understand why differences in head dynamics occur by relating these variations to the physical properties of the

dikes. Furthermore, the potential impact of climate change on the head statistics is quantified, indicating how flood risks in Dutch polders are expected to change in the future.

## 2. Study area and data

### 2.1. Dutch canal dike system

In the Netherlands, the threat of flooding is controlled by a system of flood defences, where distinction can be made between primary and regional flood defences. The primary flood defences are located along major bodies of water, such as the sea, the major rivers and large lakes, often referred to as outside waters, while regional defences are found along inland waters,

including drainage canals, man-made lakes and smaller rivers. In general, a breach in regional defences will have a smaller impact than a breach in the primary defences, though it can still have considerable consequences, as shown in Fig. 2. This study focuses on a subset of the regional flood defences, namely the canal dikes. The canal dikes are primarily located in the Western and Northern parts of the Netherlands, where the polders are located (see Fig. 2). These cultivated lowland areas serve as agricultural land as well as for human settlement. Many cities, villages and small communities are situated

throughout the polders. The water inside the polder is separated from the outside water by the primary flood defences, and the polder drainage systems manage the water inside the primary flood defences. The water is managed by discharging or pumping the polder water into canals (also called the *boezem* in Dutch), after which the canals release the water into the outside water, either naturally or using pumps (Steenbergen et al., 2009). Water levels in the canals are higher than the polder levels, resulting in a flood hazard for the polders that are protected by the canal dikes. The subsurface of canal dikes is

characterized by low-permeable soils that mainly consist of clay and peat and in the past many canal dikes breached (Van Baars & van Kempen, 2009).

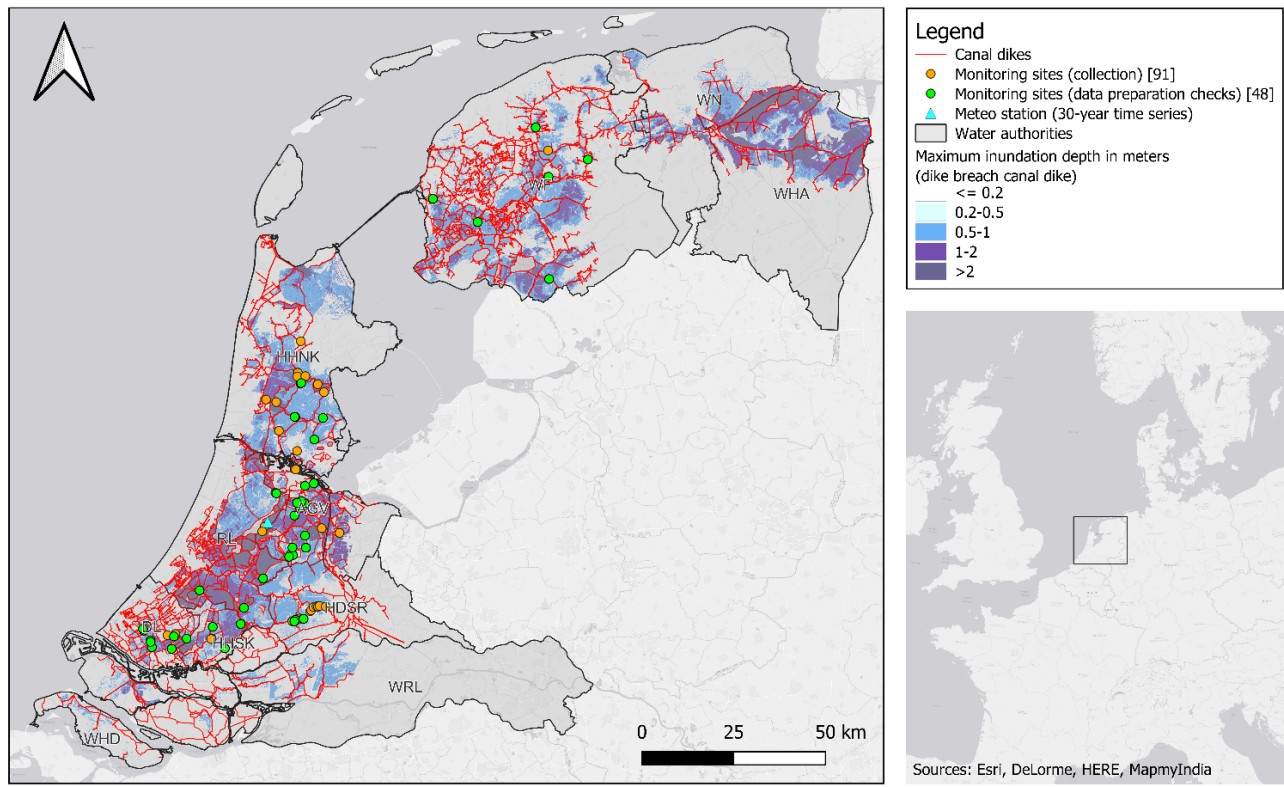

**Figure 2 Overview of the study area with the canal dike system (subset within the regional flood defence system that are located in the provinces South-Holland, North-Holland, Friesland, Groningen and Utrecht). Monitoring sites are indicated by circle markers, with orange and green denoting collected data and the data checked and utilized for further analysis, respectively. Maximum inundation depths are depicted to illustrate potential flood impact in the polders.**

## 2.2. Data collection

### 2.2.1. Head observations and preparation checks

To set-up and calibrate groundwater models, head observations in canal dikes were collected and received from seven Dutch regional water authorities, namely Hoogheemraadschap Schieland & de Krimpenerwaard (HHSK), Hoogheemraadschap Delfland (DL), Hoogheemraadschap Rijnland (RL), Hoogheemraadschap Hollands Noorderkwartier (HHNK), Weterksip Fryslân (WF), Waterschap Amstel, Gooi and Vecht (AGV) and Hoogheemraadschap De Stichtse Rijnlanden (HDSR). The crest levels of the dikes with head observations vary from around NAP+1,5 m to NAP-2,5 m (all elevations are relative to the Dutch reference level called NAP, which is approximately mean sea-level) and polder water levels ranging from NAP-2 m to NAP-6.5 m, highlighting that many canal dikes lie below sea level.

In total, 258 head observations at 91 monitoring sites were collected in this study (see Fig. 2). Multiple head observations can be located at one monitoring site, where piezometers are aligned within a dike cross-section, for example measurements

in the crest, inner slope and toe of the dike. At each site, up to five piezometers were installed with filter depths reaching up to approximately 5 meters below the surface to measure phreatic head levels. Monitoring sites can be located near each other, for example 20m, while still exhibiting varying responses due to differences in subsurface materials, highlighting the large spatial heterogeneity. Consequently, the distance between monitoring sites was not used to exclude any observations. The heads were measured with automatic pressure loggers, with hourly measurement intervals in the period between 2006

and 2023 and were resampled to daily mean values for the analysis. For further analysis, only time series that are longer than 2.5 years were selected. Additionally, any time series exhibiting visual anomalies attributed to failing measurement devices or odd behaviour such as pronounced drift, absence of fluctuations, or inexplicable jumps were removed from the dataset. The monitoring sites were also checked whether the head observations are located in a dike, since sometimes the dike is more like a quay without a slope. Ín these situations, the dike was not considered for further analyses. The head dynamics in

a dike are complex and also location-dependent within the dike profile, since the head in the outer crest can respond differently from head levels in the inner slope. Only head observations that are located in the talud zone or mid-slope of the dike were used (see Fig. 3). This is the area between the inner crest (the top of the dike at the polder side) and the toe of the dike, where the most variations in groundwater levels are expected. This is because it is farthest from the regulated water levels in the canal and polder, which are maintained at the target level.


The resulting dataset consists of 108 head time series at 48 monitoring sites, consisting of phreatic head levels measured in the dike body. The length of the head time series varies between 3 and 9 years, with an average length of 5 years.

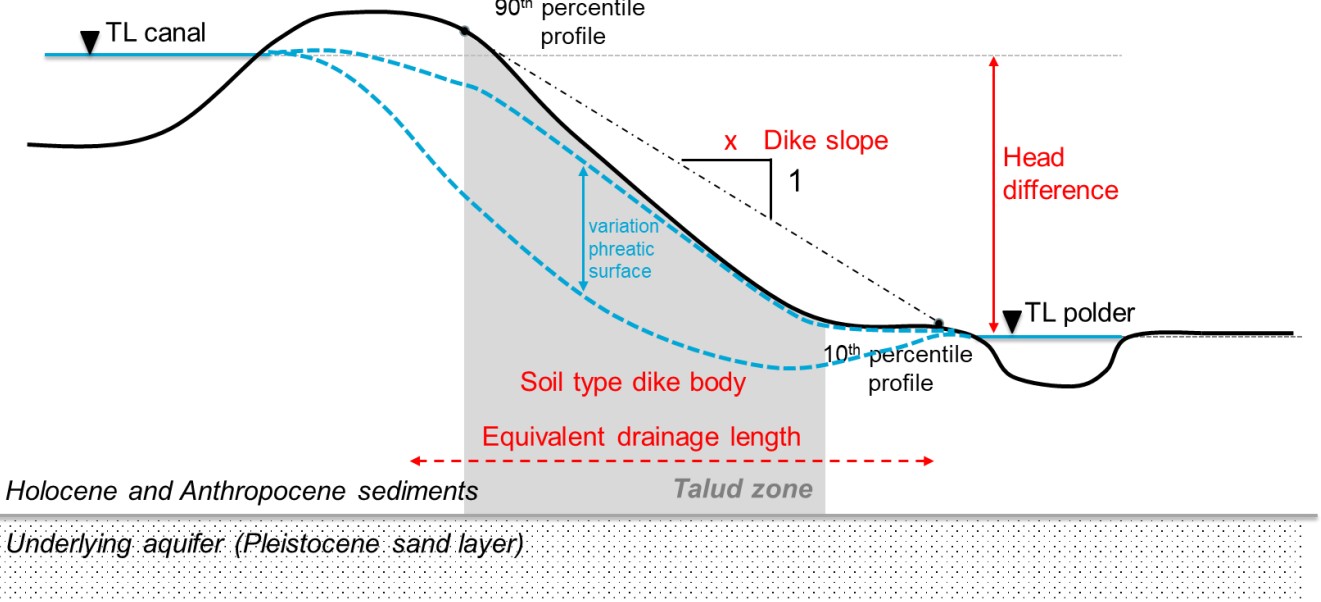

**Figure 3 A simplified cross-sectional profile of a canal dike with regulated water levels (TL=target level) on both sides (canal and**
**polder) and the variation in the phreatic surface (blue dashed lines). In this study, the groundwater levels in the talud zone of the**

**dike are analysed. The conceptualization of several dike characteristics is highlighted in red, as discussed in Section 3.3. The 90th and 10th percentile profile points were derived from the elevation profile as an approximation of the dike slope.**

### 2.2.2. Precipitation and evaporation

The Dutch meteorological institute KNMI provides several data products about weather and climate. In this study, two data products for rainfall and evaporation (on daily basis) were used, serving the purpose of 1) getting the best estimate of the historic local weather conditions at the head observation sites (for the calibration of models) and 2) getting long term time series of the weather representing the current and future climate situation in the Netherlands (for extending the head time series). First, the local historic weather conditions were extracted from rainfall and potential evaporation maps of the Netherlands, namely the RADAR-derived precipitation amounts (Wolters et al. 2023) and inverse distance weighting (IDW) interpolated potential evaporation amounts based on the KNMI ground stations. The evaporation maps give estimates of the daily Makkink reference evapotranspiration derived from ground observations of the global radiation and the average daily temperature (De Bruin, 1987). Secondly, to derive extended head time series encompassing more extreme events, 30-year of precipitation and evaporation time series corresponding to different (current and future) climate scenarios were used (Van Dorland et al., 2023). These time series are a representation of the climate and not an estimate of the actual weather. Since this study focuses on variations in peak heads caused by different head responses, rather than from spatial variations in the nature of the load (dimensions of weather events), the 30-year time series at one location was used for all sites. The station Aalsmeer, close to Amsterdam, was used which lies rather central in the Western Netherlands (see Fig. 2). In total nine 30-year time series were used. One time series corresponds to the current climate, and eight to future climate scenarios with combinations of two time horizons (2050 and 2100), two greenhouse gas emission pathways, and two types of regional climate responses. The emission scenarios include SSP1-2.6 (a low-emission scenario that assumes sustainable development) and SSP5-8.5 (a high-emission scenario that assumes fossil-fuel-intensive development). Each emission scenario is split into a wet-trending and a dry-trending regional response, reflecting the uncertainty in how precipitation patterns may shift in the Netherlands. All future scenarios with different emission levels and regional climate responses predict an increase in winter precipitation and drier summers, accompanied by increased evaporation and reduced precipitation. Although these trends occur across all future scenarios, their intensity varies across different scenarios. These combinations capture a wide range of future scenarios, allowing us to assess the sensitivity of head statistics to climate change across time, emission pathways and regional climate responses.

## 3. Method

This study takes a novel approach by combining a unique nationwide dataset of head observations in canal dikes with time series modelling to investigate how canal dikes respond to heavy rainfall. The approach developed for assessing the dynamics of hydraulic heads is shown in Fig. 4. After collecting head observations in canal dikes, time series models were set up to simulate hydraulic heads in canal dikes, using precipitation and potential evaporation as the explanatory time series. Several model structures were evaluated, and the one with the overall best performance was selected. Only models that meet the specified reliability criteria (minimum goodness of fit and sufficient time series length compared to model parameters) were selected, resulting in a set of models that explain the fluctuations of the observed head levels in different dikes with a variety of head responses (step 1). These models were forced with 30 years of precipitation and evaporation time series, corresponding to different climate scenarios (current and future conditions). This was done to obtain extended head time series that encompass more extreme events (step 2), facilitating the analysis of both the dynamics and statistics associated with extreme occurrences. The variation in head responses was quantified by analysing the coincidence of the head peaks across canal dikes, selected using the Peaks-Over-Threshold method, and classifying different clusters of dikes with similar head responses. The variation and similarity of head responses were related to several physical dike characteristics, like subsurface material and dike profile, to search for explanations of the differences found (step 3). Finally, a generalized Pareto distribution (GPD) is fitted to the head peaks that describe the probability of occurrence of peak values. The variations Wand properties of the head statistics are analysed, as well as the impact of climate change (step 4).

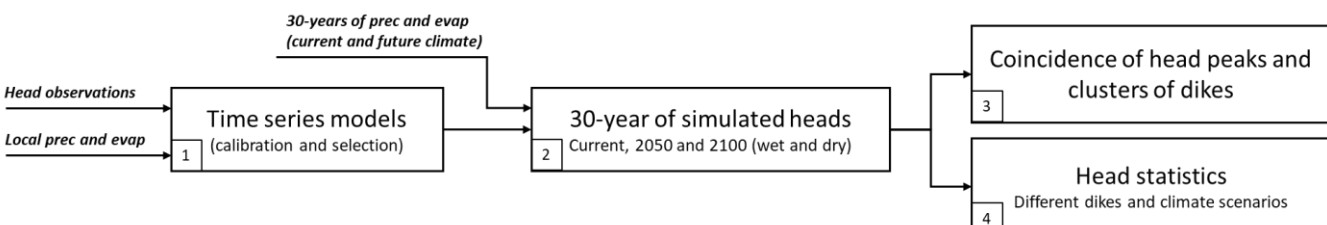

**Figure 4 An overview of the approach and its accompanying steps.**

### 3.1. Groundwater modelling in dikes

Several modelling approaches can be used to model the hydraulic head in canal dikes. Commonly used approaches comprise numerical groundwater models, like Hydrus-2D, PlaxFlow and MODFLOW, that are solutions to (systems of) differential equations that describe the flow of groundwater (Šimůnek et al., 1999; McDonald & Harbaugh, 2003). These approaches need detailed information on material behaviour for both unsaturated and saturated soils. In the case of canal dikes, Van Esch (2012) showed that it remains difficult to reproduce observed hydraulic heads in dikes, because of the uncertain conceptualization of the subsurface, spatial heterogeneity, and applied boundary conditions. Time series modelling is a simplified and abstract representation of head fluctuations at one point resulting from the complex 3D movement of water in

the dike (Bakker & Schaars, 2019). It is a data-driven approach that can estimate the contribution of independent drivers (rainfall, evaporation, water levels, etc.) on the observed head levels derived exclusively from observed data. This approach is used in this study.

### 3.1.1. Time series models

The basic principles of time series analysis comes from the statistical sciences (Box & Jenkins, 1970). Transfer function noise (TFN) modelling is a subfield within time series analysis that aims to convert one or more input series into an output series using a statistical model. Von Asmuth et al. (2002) presented a novel form of TFN models that relies on the concepts of convolution and predefined impulse response functions and is used for many applications within groundwater science. Predefined response functions are used to estimate the effect of an unit pulse of a driver, like precipitation, on the head

response. The head response is simulated through the convolution of various drivers with their response functions. The basic model structure of a TFN model to simulate heads may be written as:

$$h(t) = \sum_{m=1}^{M} h_m(t) + d + r(t)$$

where $h(t)$ is the observed heads, $h_m(t)$ is the contribution of drivers m to the head, $d$ is the base elevation of the model, and $r(t)$ are the residuals. The number of drivers $M$ in each model varies based on the selected model structure. The contribution

of driver $m$ to the head is computed through convolution:

$$h_m(t) = \int_{-\infty}^{t} S_m(\tau)\theta_m(t-\tau)d\tau$$

Where $S_m(\tau)$ is a time series of driver $m$ at preceding time $\tau$, and $\theta_m(t-\tau)$ is the associated impulse response function determining how much of that past driver still influences the head at time $t$. A variety of impulse response functions can be used to simulate the effects of certain drivers, where commonly used impulse response functions are the scaled Gamma and

the exponential response function (Collenteur et al., 2019). The exponential response function is the simplest response function with only two parameters and may be used for drivers that have an immediate effect on the head, like the shallow head levels in the canal dikes (up to a few meters below ground level). Together with the small geohydrological dimensions (the dimensions of the dike are typically only a few tens of meters), a relatively rapid response is expected, which is confirmed by measurements where head observations respond quickly to rainfall, despite the presence of low-permeable

soils.

### 3.1.2. Various time series model structures

The head fluctuations are explained by the contribution of various hydrological drivers that are convoluted with a response function. Various model structures can be chosen, incorporating different drivers with different response functions. A driver

can also be the combination of multiple drivers. The net groundwater recharge $R(t)$ is frequently used as a driver that is derived from rainfall $P(t)$ and Makkink reference evaporation $E_p(t)$ as inputs (e.g., von Asmuth et al. 2008):

$$R(T) = P(t) - fE_p(t)$$

Where the parameter f is the so-called evaporation factor used to scale the reference evaporation. This model is referred to as a linear recharge model, after which $S_m$ is substituted by the net recharge $R(t)$ and then convoluted with a response function to determine the impact of recharge on the head. In this formulation, processes such as surface runoff are not accounted for but may be relevant for canal dikes and can be incorporated using nonlinear models.

Non-linear recharge models, such as those based on soil-water balance concepts, like FlexModel or Berendrecht, offer a way to incorporate additional hydrological processes, like surface runoff. They introduce more complexity and typically require more model parameters (Collenteur et al., 2021; Berendrecht et al., 2006). Furthermore, these models can account for the nonlinear response of the head to precipitation and evaporation by using connecting reservoirs, such as interception and root zone reservoirs, which can include short-term water retention in the soil. Another non-linear model structure is the Threshold autoregressive self-exciting open-loop (TARSO) model (Knotters and Gooijer, 1999). This structure consists of two regimes (upper and lower), which are separated by a threshold. Each regime has its own exponential response function with corresponding drainage levels, but only when the head reaches the upper drainage level, the upper response function becomes active. Therefore, this model can be useful when the head response is different above a certain head level. This threshold value is not fixed but is estimated during the calibration process, just like other parameters.

The head time series were modelled using time series models as implemented in the Python open-source package Pastas (version 1.6.0) (Collenteur et al., 2019). Time series models were set up, assuming that the head dynamics in canal dikes are primarily influenced by rainfall and evaporation, and only these two drivers are included in the model. This assumption is supported by the observation that canal water levels fluctuated minimally (on the order of tens of centimeters) during the measurement period, while the observed average head range was more than one meter. Additionally, the models demonstrated an overall good fit. In total, four different model structures were employed (see Table 1). The model structure with the highest averaged goodness-of-fit across all models was used for further analysis. Selecting a single model structure ensures consistent comparison across different locations and simplifies the interpretation of results.

**Table 1 Model structures employed in the study and their characteristics, like type of recharge model, impulse response function and number of fitted parameters.**

|  | Recharge | Impulse response function | Number of fitted parameters |
|---|---|---|---|
| Linear – Exp | Linear | Exponential | 4 |
| Linear – Gamma | Linear | Gamma | 5 |
| Flex-model | Non-linear | Gamma | 7 |
| TARSO-model | Non-linear | Exponential | 7 |

### 3.1.3. Model calibration and selection

The time series models were used to characterize and simulate the heads of canal dikes with a single deterministic parameter set. For every head time series, time series models were set up, where the full series were used for calibration to maximize data utilization. This was done because the length of the available time series was limited and to avoid the missing information of the head response when splitting up the data for model calibration and model validation. Although validation tests of the models can indicate that the models are performing well and are adequate to achieve good quality model predictions in post-validation model application, previous studies showed that the most robust models are achieved when all data are used for calibration (Shen et al., 2022; Arsenault et al., 2018), which is in line with the goal of this study. Overfitting is mitigated by employing time series models with up to 7 parameters, and using head calibration data with more than 1000 data points. The model parameters were estimated using the least squares method, employing a warmup period of 10 years and without incorporating a noise model to represent the residuals. After choosing the best model structure, the calibrated models of that structure were evaluated using two criteria. These two criteria were used to determine whether a model is reliable for further analysis:

- Goodness-of-fit: the model's goodness of fit, measured by the R-squared ($r^2$), must be equal to or greater than 0.7 in the calibration period, indicating a minimum acceptable level of fit. This is also known as the coefficient of determination which is a measure of how well observed outcomes are explained by the model. The sensitivity to threshold selection was tested using threshold values of 0.6 and 0.8. Although the number of reliable models changed, both thresholds resulted in models with similar peak head responses within comparable limits. Therefore, our findings are robust to the exact choice of threshold.

- Response time: the 95% response time, the time it takes for 95% of the influence of an impulse (groundwater recharge) to dissipate, must not exceed the length of the measurement series. Time series should be long enough to cover the head response in order to estimate parameters accurately (Knotters & van Walsum, 1997). This criterion eliminates models for which the time series data isn't long enough considering the estimated model parameters.

When there are multiple reliable models of head time series available at one monitoring location, the model that provided the best fit was selected as the representative model for that location.

### 3.2. Peak selection and extreme value analysis

Peaks in hydraulic heads often occur in groups over time: an extremely high hydraulic head is likely to be followed by another since the groundwater system within dikes contains autocorrelation or memory. For extreme value analyses, we are interested in independent peaks to avoid biases and underestimation of the variability of extremes. Therefore, peaks were filtered out of the time series such that the peaks were mutually independent from each other in time by using the peaks-over-threshold (POT) method in combination with a time window. The POT method was applied with a threshold set at the

90th percentile of the analysed head series and a time window of 30 days was chosen to guarantee the selection of independent peaks. Afterwards, only the $n$ highest peaks were selected, where $n$ corresponds to the length of the time series in years, ensuring that only the most extreme values were included.

Next, a generalized Pareto distribution (GPD) was fitted to the peaks that describes the annual probability of occurrence of peak values. The GPD has three main forms, the Type I, Type II, and Type III distributions, which differ in the number of parameters and the flexibility of the tail behaviour. The cumulative distribution functions of the GPD are defined by:

$$f(x) = \begin{cases} 1 - \left(1 + \left(\xi\dfrac{x - \mu}{\sigma}\right)\right)^{-\frac{1}{\xi}}, & \xi \neq 0 \\ 1 - \exp\left(-\dfrac{x - \mu}{\sigma}\right), & \xi = 0 \end{cases}$$

where x is the hydraulic head and the three parameters of the GPD are called the scale ($\sigma$), shape ($\xi$) and location ($\mu$)
parameters. When $\xi=0$ the GPD is equivalent to the exponential distribution. All generalized Pareto distributions were explored in a sensitivity analysis. In the case of peak hydraulic heads, the exponential distribution was preferred due to its relative simplicity and its suitability for the process, as suggested by visual inspections of the tail of the distribution and the peak values. One key characteristic of the extreme value distribution is the decimate height. It is defined as the increase in head level that occurs when the return period increases by a factor of ten, and is a relevant characteristic in load and dike
safety analyses (Wojciechowska, 2015; Schweckendiek, 2014).

The effects of climate change were quantified by estimating the extreme value distribution, or head statistics, of simulated head time series for different time horizons and climate scenarios (including emission scenario and regional response). To compare future scenarios with the current climate, the increases or decreases in head levels at a certain return period were
305 compared, and the so-called probability factor was used. This metric expresses how the frequency of a head level with a 100-year return period is expected to change under climate scenarios. A probability factor of five implies that a head level that occurs, on average, once every 100 years in the current climate occurs once every 20 years under future climate conditions.

### 3.3. Clustering head responses and relating to physical dike characteristics

The relationships between the head responses and several physical dike characteristics were examined. These characteristics
include the subsurface material of the dike body, the head difference (water level difference between the target levels of the canal and polder), the dike slope and the equivalent drainage length (see Fig. 3). Dike slopes can be seen as a measure of the hydraulic gradient within the dike body, which influences the horizontal groundwater flow and the head response. In addition, steep dike slopes can reduce recharge, as they can increase surface runoff that limits the possibility for water to infiltrate into the dike. The slope of a canal dike was not always obvious, since the dikes don't always have a typical
geometry, where the profile is irregular and the crest, the berm and the toe of the dike are not clearly recognisable. The dike

slope was obtained from an elevation map of the Netherlands with a horizontal resolution of 0.5m (Actueel Hoogtebestand Nederland, 2022), and was calculated by referencing two percentile points on the elevation profile (10th and 90th percentile), as an approximation of the slope between the crest and the toe of the dike. The target levels of the canal and polder were obtained from the local water authorities, which were used to calculate the head difference. The equivalent drainage length was determined by dividing the head difference by the estimated dike slope, providing an estimate of the horizontal distance over which water is effectively drained from the dike. The subsurface of the dike was based on borehole descriptions or Cone Penetration Testing (CPTs) at the monitoring sites. In the absence of soil investigation, a detailed three-dimensional model (GeoTOP) of the upper 30 meters of the subsurface of the Netherlands was used (Stafleu et al., 2012).

### 3.3.1. Clustering head responses

Differences in peak head behaviours were examined by analysing the coincidence of selected head peaks across all dikes in the 30-year simulated head time series. These simulations were based on 30 years of rainfall and evaporation at the same KNMI station to isolate the effect of differences in peak heads from the spatial variability of weather events. If peak heads at two different dikes occurred on the same day, they were assumed to coincide. By calculating the percentage of coinciding peak levels for each dike pair, a coincidence matrix was formed. This matrix provided a quantitative measure of how often peak heads align across different dikes, indicating their response to similar weather events. Based on this matrix, dike clusters were identified. These clusters consist of dikes where peak heads were driven by similar weather events, while dikes in different clusters experienced peak heads caused by distinct weather events. The clusters are estimated using the k-means clustering algorithm (Hartigan & Wong, 1979), where the number of clusters (k) has to be given beforehand and is based on the mean Silhouette score of all samples (Rousseeuw, 1987) and the "elbow method", as implemented by Yellowbrick (Bengfort & Bilbro, 2019).

### 3.3.2. Statistical tests

The relationships between these physical dike characteristics and characteristics of the impulse response functions, as well as clusters of dikes, were examined. Various statistical tests were employed to assess these relationships by calculating the p-value for different types of variables, both categorical and continuous; Wald Test for comparing two continuous variables, the Wald Chi-Squared Test is used for two categorical variables, and the Kruskal-Wallis Test is employed for one continuous and one categorical variable. The analyses of relationships with subsurface materials were limited to clayey and peaty dikes, as the dataset includes only one sand dike and Dutch canal dikes are generally composed of these materials.

## 4.   Results

### 4.1. Modelled head responses in canal dikes

#### 4.1.1.   Selecting reliable time series models

For each of the 108 head time series across 48 monitoring sites, time series models with various model structures were created, calibrated and evaluated. To illustrate the performance of different model structures, Fig. 5 gives an example for the monitoring site at Molenlaan (site DL4). The linear models (exponential and gamma response functions) are not able to model the head response for the full range of head levels, as can be seen in the scatterplot. For this location, it appeared that the head response was nonlinear. Both the TARSO and Flex models provide a better fit across the entire range of head levels. Notably, the TARSO model shows improved performance in capturing head levels at the most extreme ends of the range.

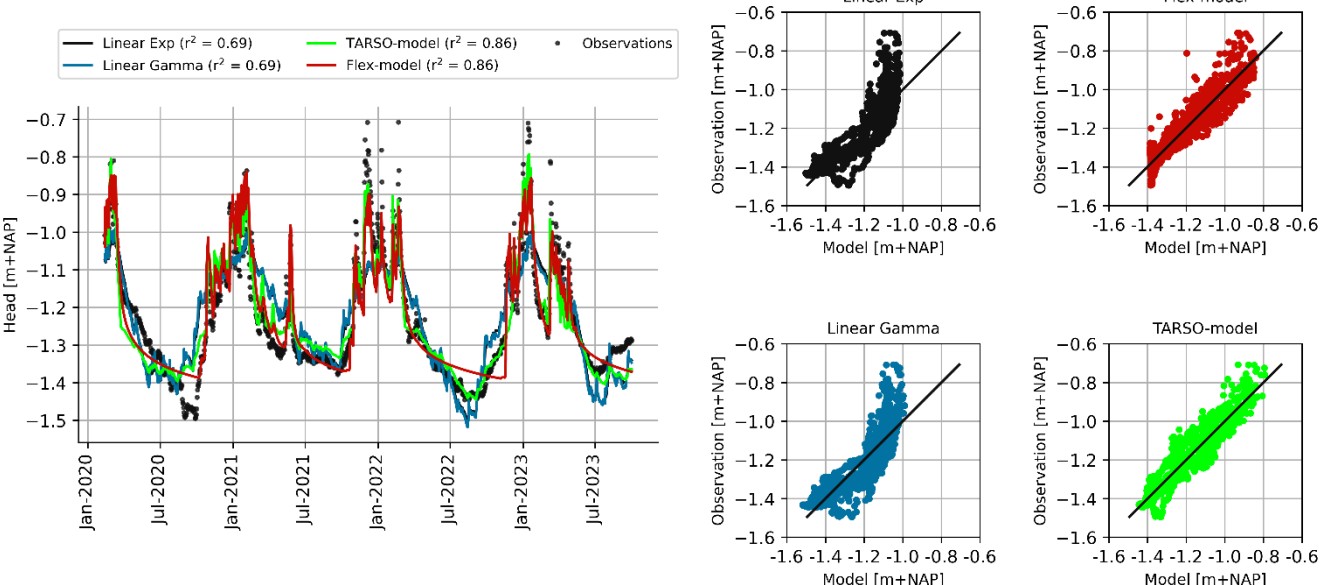

**Figure 5 The performance of different model structures at Molenlaan (DL4). The left graph shows the simulated and observed head levels and on the right four scatterplots of the model structures showing the observed and simulated daily head levels (the black line indicates the 1:1 line).**

The TARSO-model demonstrated the best performance among all the calibrated models. It has the highest average goodness-of-fit (an average $r^2$ of 0.74) and performed as the best structure for 81% of the models. The second-best model was the linear recharge model with the Gamma response function (an average $r^2$ of 0.68), while the Flex model structure performed, on average, the worst with an average $r^2$ of 0.63. However, the Flex model still performed best for 12% of the models, indicating that in some cases, a more detailed nonlinear representation of recharge processes can be beneficial. Overall, the head dynamics in Dutch canal dikes can be best modelled with a non-linear structure that incorporates two

regimes, which are separated by a threshold. This non-linear behaviour can be the result of various soil layers in the dike body, each with distinct hydraulic properties, and changes in infiltration rates or nonconstant storage capacities of the

365 unsaturated zone during the dry season (Knotters & de Gooijer, 1999). Next, the calibrated TARSO models were evaluated using the reliability criteria (goodness-of-fit and response time) and for every monitoring site, the model that met the reliability criteria and had the highest goodness-of-fit score was selected. At 38 out of 48 monitoring sites (79%) models were developed with $r^2$ of 0.7 or larger. The other reliability criterion, that the 95% response time should not exceed the length of the measurement series, reduced the number of sites with reliable models to 35 (73%). In Fig A1 in the

370 Appendices, detailed information is provided on the best-performing TARSO models at each monitoring site, including the r² values and corresponding response times. In addition, plots of the measured and simulated heads during the measuring period for all selected monitoring sites are shown in Fig. A2 in the Appendices. This set of models was used for further analysis.

The model results of one of the selected time series models are shown in Fig. 6. The simulated heads closely match the observations with a $r^2$ of 0.84, indicating a close one-to-one relationship between simulations and observations. The model overestimates the heads in the summer of 2019, which was particularly dry in terms of head levels. These differences may be due to inaccurate precipitation data used in the model (as summer precipitation can be highly localized) or disturbances during installation, affecting the head levels in 2019. The other dry summer in 2022 was captured more accurately. The

calibrated block response functions for both the upper and lower regimes are shown in the lower right graph in Fig. 6. These functions are dimensionless and represent the unit head response to a 1-day recharge event of 1mm. The actual head change is obtained by scaling this response with the recharge input. Two key characteristics of these functions are 1) the peak of the block response and 2) the 95% response time. The peak of the block response represents the maximum increase in head level that would occur. The 95% response time, further on referred to as the response time, is a measure of the memory of the

groundwater system and, in this case, represents the time it takes for 95% of the influence of an impulse (groundwater recharge) to dissipate. In the TARSO models, the upper and lower regimes are separated by a fitted threshold parameter, with head responses differing in each regime. For this model, the head response in the upper regime reacts more strongly to recharge events (higher peak block response) and dissipates more quickly (lower response time) than the lower regime. Whether this behaviour is consistent across all dikes is examined in the next subsection.

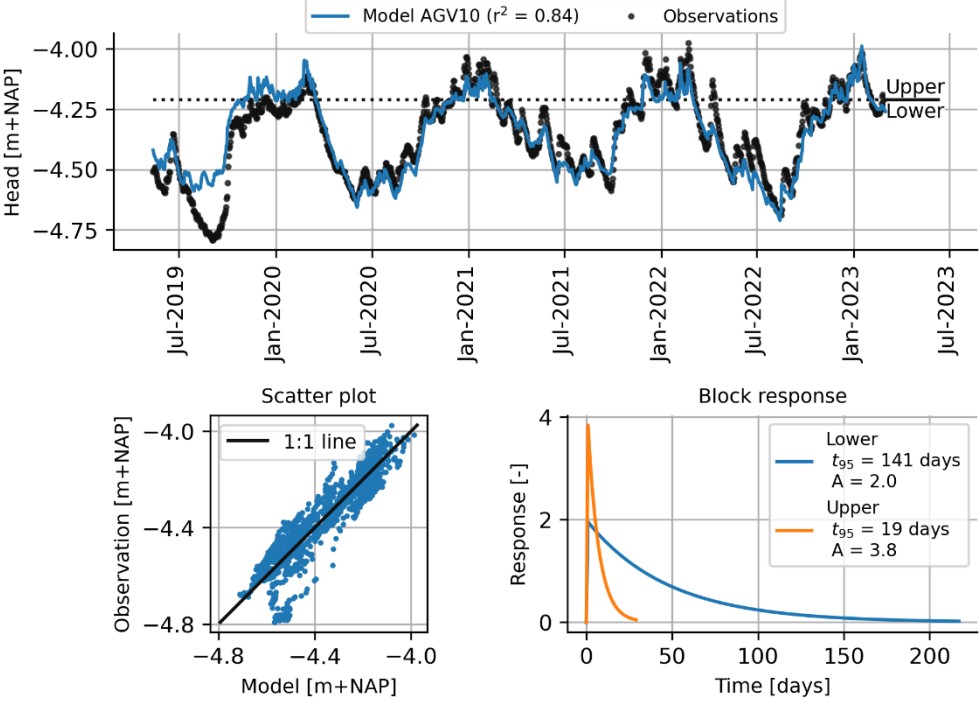

**Figure 6 Illustration of the simulated head levels with the time series model in comparison with the observations for dike AGV10. The graph on top shows the observations and model series where also the upper and lower regime are indicated with the black dotted line and the bottom left shows the scatter plot of observed and simulated heads. The lower right graph shows the impulse response functions of the upper and lower regimes, including the parameters of the peak of the block response (*A*) and the response time (*t₉₅*) in the legend.**

### 4.1.2. Characteristics of impulse response functions

The head dynamics of various dikes were quantified by examining the two key characteristics of the impulse response functions, namely the peaks of the block responses and the response times. Figure 7 shows these characteristics for both the upper and lower regime of the TARSO-model, where the colors indicate in which water authority region the dikes are located. The colors initially appear random, indicating no clear spatial pattern in the variation of head responses across regions, likely due to the heterogeneous subsoil conditions and other dike characteristics within the canal dike system. The response times of the upper regimes of the dikes are generally short, mostly ranging from about 2 to 50 days, with two exceptions where response times exceed 250 days. These two exceptions have a very small peak of the block response compared to other dikes, which have a large variation in the peak response of the upper regime, with values reaching up to nearly 15. This variation may result from different soil storage capacities and the redistribution of infiltrated water within the dike, which can accumulate in the talud zone causing large head responses. The response times of the lower regime show more variation than those of the upper regime, with most dikes ranging between 100 and 600 days. Meanwhile, the peak block for the majority is below 5.

Two key patterns of non-linearity appear in nearly all locations when analysing the response functions. First, the response time is longer for the lower regime than for the upper regime (34 out of 35 sites). Second, the peak of the block response is higher in the upper regime than in the lower regime (32 out of 35 sites). These observed differences may be explained by underlying physical processes. For example, longer response times in the lower regime can be caused by head gradients (as a driver of groundwater flow) that depend on the head level itself, the presence of various soil layers with different permeabilities, or the fact that head levels closer to the surface increase the degree of water saturation which affects the hydraulic conductivity and response times in a non-linear way. The lower peak of the block response for the lower regime can be caused by non-linear processes in the unsaturated zone, where lower head levels generally allow more water storage, and root water uptake further increases storage capacity (Berendrecht, 2006). In contrast, when head levels are higher, capillary action draws water upward from the saturated zone, increasing moisture in the unsaturated zone and reducing the amount of air-filled pore space. This limits the potential for additional water storage.

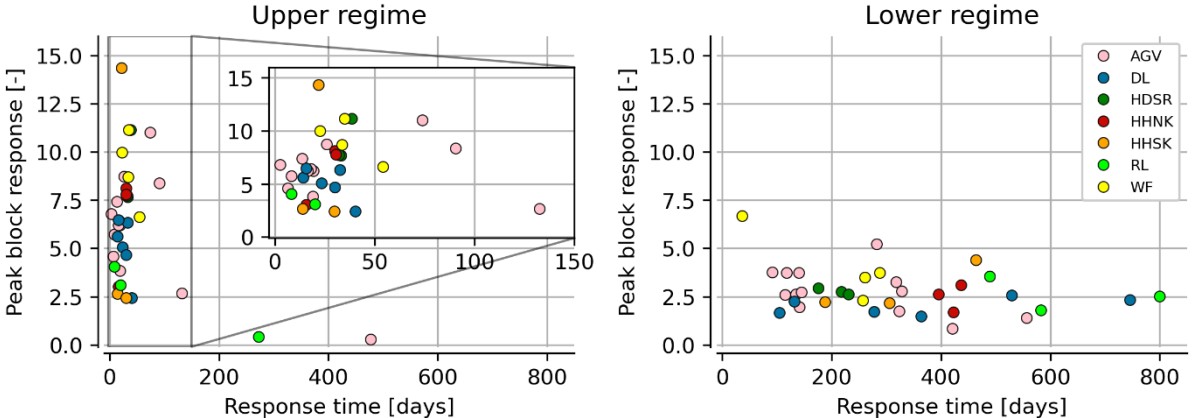

**Figure 7 The characteristics of the impulse response functions at 33 canal dikes (peak of the block response and 95% response time) for the upper and lower regimes, where the colors indicate the various water authority regions where the dikes are located. The inset at the upper regime shows a zoomed-in view.**

### 4.2. Variation in peak head responses

#### 4.2.1. Coincidence of head peaks and dike clusters

The coincidence matrix, shown in the left graph in Fig. 8, describes how often peak heads occurred on the same day across different dikes in the 30-year simulated head time series. This illustrates whether peak heads tend to result from the same or different weather events. Based on this matrix, the elbow method identified an optimal number of clusters at k=4, while the Silhouette score indicates that either k=2 or k=4 could be optimal, with scores of 0.442 and 0.439, respectively. Therefore, the number of clusters that are identified using the k-means clustering algorithm was set at four. The resulting clusters of dikes are called clusters A, B, C and D, and are indicated within the coincidence matrix, see Fig. 8. While the three clusters

435 A, B and C are less distinct from each other with still moderately high percentages of coincident peaks, sometimes exceeding 50%, cluster D has a very distinct peak behaviour. This cluster consists of only two dikes of which the time series models have deviant impulse response functions; longer response times and smaller peak block responses in the upper regime, compared to the other clusters (upper right graph in Fig. 8). In general, the distinctive peak behaviour between clusters is strongly influenced by the response time of the upper regime, with average response times of 16 days, 32 days and 77 days,

for clusters A, B and C, respectively. This twofold increase in response times for each cluster results in distinct rainfall events leading to peak heads. These longer response times cause peak heads to be driven by more prolonged rainfall events, resulting in peaks that typically occur later in the winter. More details on the differences in seasonality of peak heads can be found in Appendix A2, including analyses on the average timing and distribution of peak heads throughout the year.

The dike clusters do not exhibit a clear spatial pattern, as shown in Fig. A4 in the Appendices. In some regions, such as within the water authority AGV and HDSR, dikes appear to be mainly in clusters A and B, respectively, while in other regions, no clear spatial pattern is observed. Although canal dikes within the same polder may have similar dike characteristics, there can still be large spatial variabilities of those characteristics across a region (see Fig. A6 to A9 in the Appendices). Moreover, even dikes that initially appear to have similar characteristics can exhibit different head responses,

which are examined in more detail in the following paragraph.

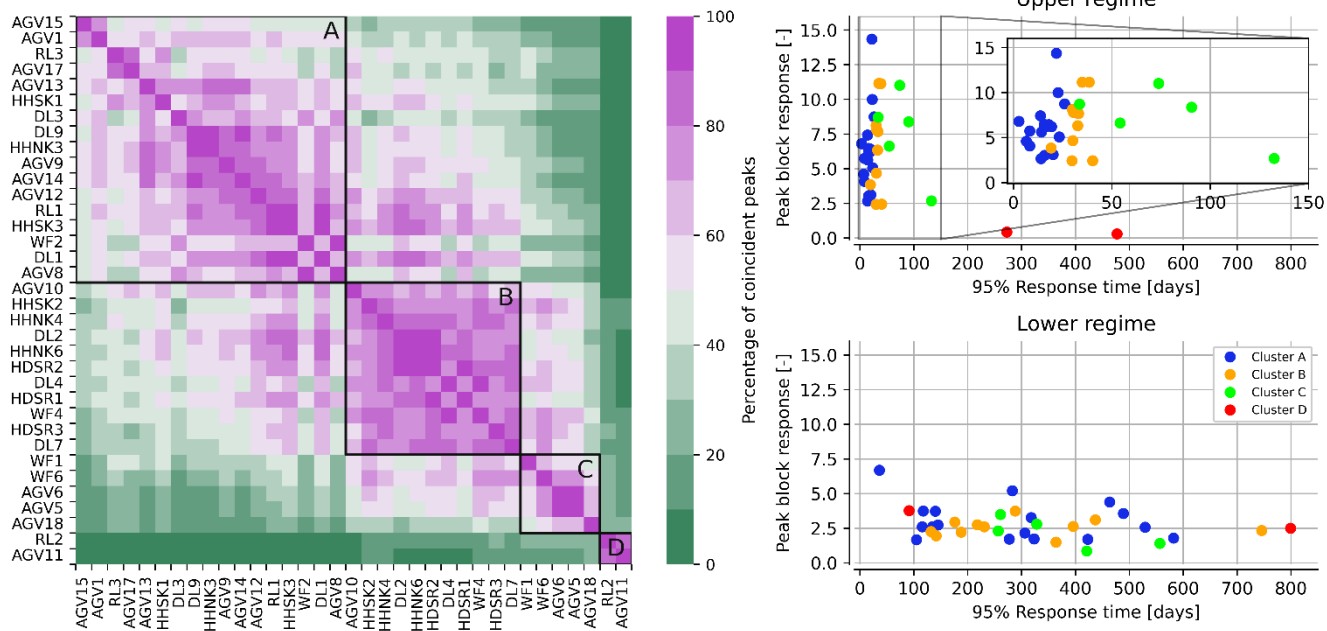

**Figure 8 The coincidence matrix of the head peaks at the canal dikes, including four identified clusters (left graph). Within every cluster, the locations are ranked based on the 95% response time. The characteristics of the impulse response functions of the dikes within every cluster are shown in the right graph.**

#### 4.2.2.  Clusters in relation to physical dike characteristics

Can differences in peak head responses be explained by physical dike characteristics? Table 2 shows the p-values for the relationships between physical dike characteristics and both the clusters and the characteristics of the impulse response functions (upper and lower regimes), indicating that most relationships were not statistically significant (p-values greater than 0.05). However, the subsurface material of the dike appears to play an important role as a distinguishing characteristic in the dike clusters, with a p-value of 0.03. Statistical tests used to calculate the p-values can be inappropriate due to violations of test assumptions, small sample sizes, multiple comparisons and data dependency, all of which can lead to misleading statistical conclusions (Greenland et al., 2016). Therefore, the physical dike characteristics within every cluster are also visually analysed and, as expected from the statistical tests, there is a large variation of dike characteristics within every cluster, see Fig. 9. The clayey and peaty dikes appear in all dike clusters, with most clayey dikes in cluster B (see also Fig. A10 in the Appendix). Furthermore, small dikes, with drainage lengths less than 20 meters and often associated with steep slopes, are found only in Clusters A and B, which are clusters with the smallest response times. Furthermore, the median equivalent drainage length of the clusters increases from Cluster A to Cluster D. This indicates the importance of dike geometry, where shorter distances from a drain (the canal or ditch) lead to faster dike drainage and smaller response times. This can be explained by the fact that the hydraulic gradient, which drives water towards the drain, increases with shorter distances. Yet, determining the cluster to which dikes with certain dike characteristics belong isn't straightforward, since all clusters include dikes with various characteristics.

**Table 2 P-values of the relationship between the considered physical dike characteristics (rows) and the clusters of dikes and the 95% response time and peak block response for both the upper and lower regime (columns).**

|  |  | Upper regime |  | Lower regime |  |
| --- | --- | --- | --- | --- | --- |
|  | Clusters | 95% response time | Peak block response | 95% response time | Peak block response |
| Subsurface material | 0.03 | 0.32 | 0.32 | 0.32 | 0.32 |
| Equiv. Drainage length | 0.19 | 0.19 | 0.31 | 0.22 | 0.39 |
| Dike slope | 0.35 | 0.64 | 0.90 | 0.96 | 1.00 |
| Head difference | 0.47 | 0.11 | 0.08 | 0.08 | 0.06 |

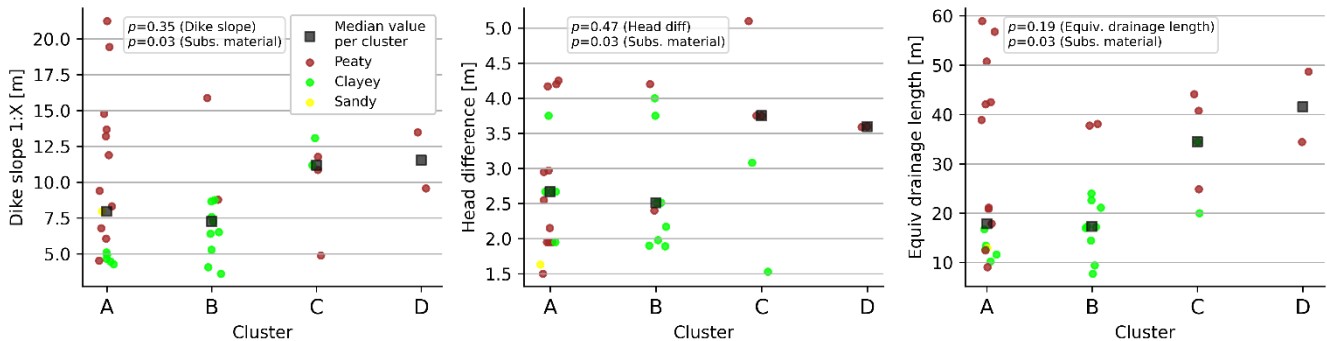

**Figure 9 The occurrence of dike characteristics across clusters (A, B, C and D), where the colors indicate the subsurface material. The median values per cluster are marked (■).**

### 4.3. Statistics of head peaks

#### 4.3.1.    Head statistics and decimate height

The selected hydraulic head peaks in the 30-years simulated head time series are used to estimate the extreme value distribution of peak head levels by fitting an exponential distribution. The statistics of hydraulic head levels at a dike along the Beemster polder (HHNK3) are used as an illustration, see Fig. 10. The decimate height at HHNK3 is approximately 7 cm, while across various dikes, the values range from around 5 to 50 cm with a median decimate height of 15 cm (as seen in

the right graph in Fig. 10). Lower decimate heights are found at dikes with smaller peak block responses in the upper regime, in combination with shorter response times. Since these characteristics of impulse response functions do not exhibit a clear relationship with dike characteristics (see Table 2) or a distinct spatial pattern, the decimate height also does not follow a spatial pattern, as shown in Fig. A11 in the appendices.

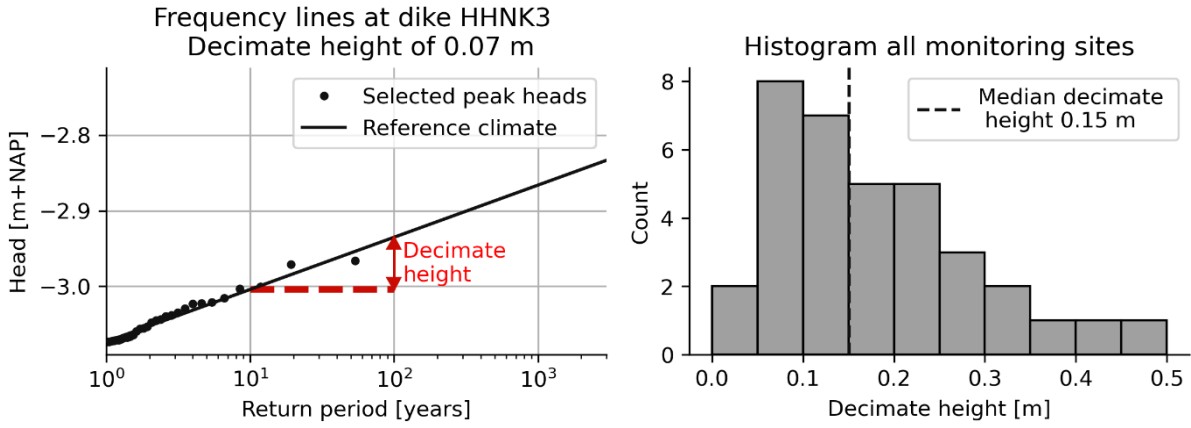

**Figure 10 Left: Frequency lines of hydraulic head levels at a dike along the Beemster polder (HHNK3) based on precipitation and evaporation corresponding to the current climate. Right: a histogram of the decimate heights of all dikes considered of which the median decimate height is 15 cm.**

### 4.3.2. Impact of climate change

For one location, the resulting frequency lines of various climate scenarios in the year 2100 are shown in Fig. 11 (left graph). For this location, all climate scenarios result in more extreme head levels or head levels to occur more frequently. In the Hw-scenario (high emissions and wet regional climate response), which has the largest increase in winter precipitation according to Van Dorland et al.(2023), the head level at a return period of 100 years increases by 10 cm. Due to the small decimate height, the original head level, occurring once every hundred years, is expected to happen once every 15 years, indicating a sixfold increase in frequency or a probability factor of six. The probability factors across all dikes, shown in the right graph in Fig. 11, range from about three times less frequent to seven times more frequent across all climate scenarios in 2100. This variation could not be directly linked to the clusters of dikes, however, it was found that dikes with longer response times in the lower regime appear to be less impacted by climate change. This can be explained by the fact that these dikes dry out more during drier summers. As a result, a more dried-out dike allows for greater water storage when rainfall returns, causing head peaks to occur less frequently. However, the impact of drier summers on the occurrence of extreme head levels is counterbalanced by wetter winters, which both occur in all climate scenarios. Therefore, the characteristics of both the head response to changing winter precipitation and summer evaporation determine the overall impact of climate change on extreme heads. Under the low-emission scenarios, changes in the frequency of extreme head levels remain small in both 2050 and 2100. This is due to relatively moderate increases in winter precipitation and summer evaporation, which appear to balance each other. Under high-emission scenarios, the impact of climate change in 2050 is, on average, moderate. However, at some dikes, under the dry regional response, the frequency of extreme heads reduced by more than a factor of three. By 2100, however, high-emission scenarios indicate an increase in the frequency of extreme head levels, caused by wetter winters, with the most significant impact observed in the wetting regional response.

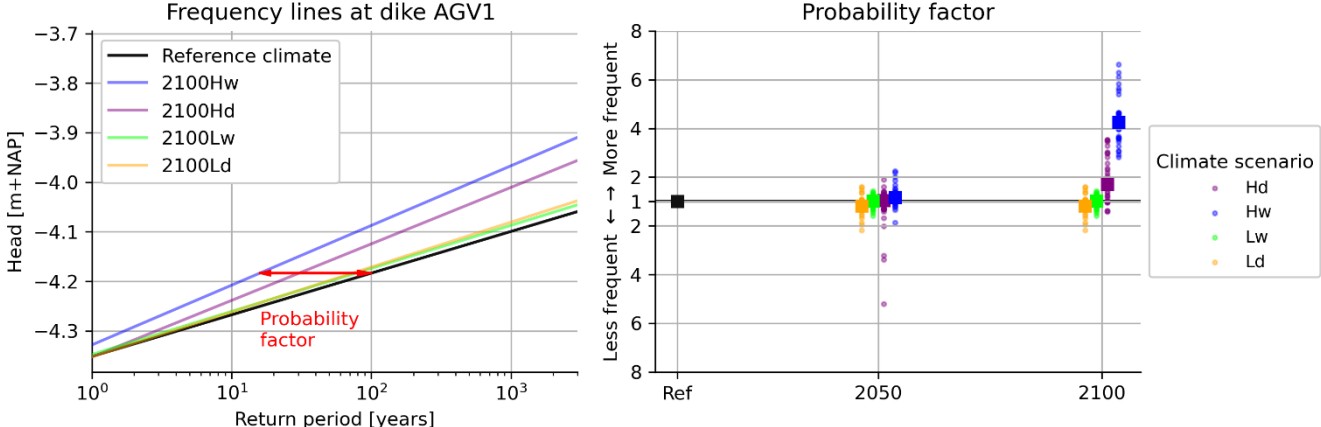

**Figure 11 Left: The frequency lines of head levels for various climate scenarios in 2100 (High/Low emissions; dry/wet regional climate response) at the dike in a polder south of Amsterdam. Right: the probability factor of all dikes (●) and the median value (■) for different time horizons and climate scenarios.**

## 5. Discussion

### 5.1. Limitations and recommendations

Given the extensive canal dike system with thousands of kilometers of canal dikes and the heterogeneity of dike bodies, the number of available observations, both in terms of locations and measurement duration, is limited. As a result, it remains uncertain to what extent this set of 35 reliable models used in this study is representative of the head responses in Dutch canal dikes. In addition to the limited number of observations, two factors further affect representativity. First, the chosen TARSO-model structure influences which head responses are included in the dataset. The reliability criteria applied in this study filtered out observations and associated head responses that could not be modelled adequately, meaning that certain head responses that did not fit the selected models were also excluded. As a consequence, the selected models may not fully capture all relevant processes influencing the head response of canal dikes. For instance, changes in hydraulic conductivity and additional non-linear processes, such as surface runoff from excessive rainfall, are not explicitly accounted for. This limitation also affects the reliability of hydraulic head estimates beyond the measured range of head variations, making them more uncertain. The model uncertainties associated with the time series models were not considered in this study. Furthermore, hypotheses and possible explanations of why the TARSO model fits the head response of canal dikes best require further research. Field measurements and 2D numerical modelling could help improve the understanding of water flow and validate the suitability of TARSO models. Second, it was assumed that each dike has a single head response, as only the best-fitting model within the talud zone was selected to represent that location. However, variations in head responses can exist within the talud zone of a single dike. At fourteen monitoring sites, multiple reliable models were available within the talud zone, allowing an assessment of this variability. At ten out of these sites, the response times of the upper regime differed by only ten days, indicating similar peak behaviour. Despite the uncertainties surrounding the representativity of this model set, this study demonstrates the value of observations and makes the data publicly available. We hope this will encourage further long-term measurement campaigns to extend the available data and improve the understanding of head responses in canal dikes.

In this study, the k-means clustering algorithm was used to cluster dikes based on the coincidence of peak heads. However, two key factors influence clustering outcomes. First, the choice of clustering method affects the results. Various clustering methods exist, and different methods may lead to different clustering results (Everitt et al., 2011). Second, the input data used for clustering determines the outcome. In this study, clustering was based on the percentages of coinciding peak heads across dikes, as this parameter best aligns with our objective of evaluating peak head variability in canal dikes. However, for different purposes, alternative parameters or datasets can be chosen to cluster dikes. While clustering provides a practical way to manage variability, in reality, there is no clear distinction or a clear-cut between different dikes in terms of head responses and peak behaviour. Instead, it is a gradual shift across a spectrum of head responses.

Next, the variation in head responses and the found clusters were linked to dike characteristics. These dike characteristics themselves were difficult to determine clearly, leading to uncertainty. For instance, the profile of the canal dike can be very irregular and in many cases there is no singular, defined slope. Additionally, canal dikes are typically made up of multiple soil types, making it difficult to classify a peaty or clayey dike. With all these uncertainties, it's challenging to define clear characteristics for dikes and identify patterns, if not impossible. The uncertainty of the subsurface material of dikes is especially large for those based on the subsurface model GeoTOP rather than borehole descriptions or CPTs. To evaluate the impact of GeoTOP data on the findings, a sensitivity analysis was conducted by testing the effect of excluding this data. While the p-values changed slightly, the overall results remained consistent, with the p-value for the relationship between subsurface material and clusters increasing to 0.05, highlighting the importance of subsurface material in the analysis. Furthermore, we examined the relationship between the head response and various dike characteristics, looking at each characteristic separately. It is possible that considering the combination of various characteristics might reveal a clearer pattern. However, it is important to keep in mind that even if there's a relationship between head responses in canal dikes and dike characteristics, it is challenging to determine the local dike characteristics of each individual dike stretch, because of the heterogeneous nature of the dike system. This heterogeneity exists in both longitudinal and cross-sectional directions. As a result, it can be expected that the head responses of the canal dikes have large spatial variations and can even differ for dikes that are close to each other.

Lastly, there are limitations in how the impacts of climate change on peak heads were modelled. This study considered the impact of climate change effects only in terms of changes in precipitation and potential evaporation, assuming that the head response itself remains unchanged. However, head responses can change over time due to shifts in hydraulic conductivity and water retention capacity of soils. These changes may result from cyclic wetting and drying of soils, leading to swelling, shrinkage, soil consolidation and alterations in dike structures over time (Stirling et al., 2021; Azizi et al., 2019). As climate change is expected to intensify dry-wet cycles, these processes may become more pronounced, potentially affecting the stability of dikes. Additionally, dike resistance can be further influenced by factors such as shear-strength reduction, soil compaction, and peat decomposition. Quantifying all the effects of future climate scenarios is challenging, as both the hydraulic and mechanical behaviour of soils are intertwined and impacted. Translating these effects into slope instability is even more complex, which is beyond the scope of this study. To better understand and quantify changes in head responses within dikes, continuous long-term monitoring, preferably exceeding 10 years, is essential.

## 5.2. Implications for dike safety

This study quantified variations in peak head responses of canal dikes, which are relevant for estimating regional or national flood risk levels in polders. Consider an imaginary canal dike ring along a small polder where each individual dike section is assumed to have a failure probability of 1/100 per year, with fully spatially correlated load and strength characteristics. The weather conditions across this polder are uniform, with equal precipitation and evaporation everywhere. Under these

conditions, the probability of flooding in the polder equals the highest failure probability of the individual dike sections. In contrast, suppose there were four different types of dikes, each with its own peak head responses, resulting in four statistically independent load conditions. If their strength characteristics remain fully spatially correlated, the flood probability in the polder increases to approximately 1/25 per year; a factor four higher than with fully spatial correlated load characteristics. In general, this can be calculated by $P_{f,sys} = 1 - \left(1 - P_{f,sec}\right)^n$, where $P_{f,sys}$ is the failure probability of the dike ring (or flood probability of the polder), $P_{f,sect}$ is the failure probability of the individual dike sections and $n$ is the number of statistically independent dike sections. The spatial correlations in loads and strength are crucial for accurately estimating flood risk levels in regions, and this study provides valuable insights into the variation of loadings in dikes by considering the different head responses that affect these estimates. Besides variations in head responses, regional risk levels are also affected by natural spatial variability in weather events. This spatial variability can increase the number of statistically independent load events, depending on the scale considered.

Gariano and Guzzetti (2016) reviewed the literature about the impact of climate change on landslides, both in natural and engineered slopes, and concluded that the risk of shallow landslides can increase (triggered by short and intense rainfall events), while the risk of deep-seated landslides may decrease or show no significant change (related to long rainfall periods). This is primarily due to changing meteorological conditions that lead to higher head levels, reducing the shear strength, soil suction and cohesion, and increasing the weight (wet density) of slope materials, all of which contribute to increasing slope instability. Deep-seated landslides appear to decrease or show no significant change, because these types of landslides depend on monthly and/or seasonal rainfall amounts. These more prolonged rainfall events are expected to decrease in regions, like the Alps (Rianna et al., 2014; Gariano and Guzzetti, 2016). Canal dikes show similar behaviour to shallow landslides, with, on average, an increased risk of instability in the future. However, the impact of climate change on canal dikes can still vary considerably, with extreme head levels projected to occur either more or less frequently depending on the specific head response. While this study could not identify definitive explanations for these differences, the results suggest that both the head response to changing winter precipitation and summer evaporation play a role in the overall impact of climate change. These insights are relevant for assessing dike safety over time and enhancing adequate dike design.

Another interesting finding for dike safety assessments is the small decimate height in the canal dike head statistics. For dikes with a decimate height of only 5 cm, yearly occurring head levels (T=1 year) are only 15 cm lower than extreme head levels that occur on average once every 1000 years. Measurements in these dikes are particularly valuable for improving safety assessments, as observed heads are closer to extreme levels, allowing for more accurate extrapolation with fewer uncertainties in modelling extreme levels (Wojciechowska, 2015). Furthermore, short-term measurements can already capture relatively high observed loads, providing valuable insights into the actual dike strength through reliability updating

techniques (Schweckendiek, 2014). Nevertheless, for dikes that are marginally stable, this small increase in head level can still be a trigger for dike failure.

620

## 6. Conclusions

This study aimed to assess the dynamics of peak heads in Dutch canal dikes at a national level by analysing variations in head responses, head statistics, and the impact of climate change. This was done using time series models calibrated on a unique dataset on head observations in dike systems, consisting of 108 head time series across 48 monitoring sites. Various model structures were evaluated and it was found that the non-linear TARSO-model outperformed the other (non-)linear models. This model consists of two regimes (upper and lower), separated by a threshold, each with its own exponential response function and drainage levels. This threshold non-linear behaviour in canal dikes can be attributed to several factors. Groundwater flow is driven by head gradients and hydraulic conductivities, which both can vary vertically and depend on the head level itself, while rising head levels near the surface non-linearly affect hydraulic conductivity and water storage in the unsaturated zone. The TARSO-model was selected to calibrate time series models for all head time series, which were then evaluated using two reliability criteria (goodness-of-fit and response time), resulting in a set of 35 reliable models.

Differences in peak head behaviour between various canal dikes were examined by analysing the coincidence of head peaks across all dikes in 30-years simulated head time series. Four clusters of dikes were identified consisting of dikes where peak heads were driven by similar weather events, while dikes in different clusters experienced peak heads caused by distinct weather events. The differentiating factor was the response times of the upper regime of these dikes, where longer response times caused peak heads to be driven by more prolonged rainfall events. The identified dike clusters do not exhibit a clear spatial pattern. The reasons are the large spatial variability of dike characteristics and the fact that even dikes with similar characteristics can exhibit different head responses. While the subsurface material and dike width appeared to be important factors influencing the variations in head responses, their presence in multiple clusters indicates that these characteristics alone do not definitively determine the head response.

Next, peak head statistics were derived across the canal dikes, revealing that the median decimate height is only 15 cm, ranging between 5 and 50 cm. This indicates that yearly occurring head levels are, on average, relatively close to extreme events. Dikes with lower decimate heights were associated with smaller peak block responses and shorter response times in the upper regime. Since these characteristics of impulse response functions do not exhibit a clear relationship with dike characteristics or a distinct spatial pattern, decimate heights also do not follow a spatial pattern. With climate change driving higher winter precipitation and summer evaporation, head statistics are changing. Results showed that head levels with a return period of 100 years are expected to occur about 3 times less frequently to 7 times more frequently by 2100, depending on the climate scenario and the type of canal dike. Dryer summers can reduce the frequency of extreme peak heads by lowering head levels during summer, which increases dike water storage capacity when rainfall returns. However, most climate scenarios project a higher frequency of extreme head levels by 2100, caused by a wetter winter trend. The varying impact of climate change on dikes is largely attributed to the response times of the lower regime. Dikes with longer response

times seem to be less affected by climate change, as they experience greater drying during drier summers. However, this increased drying during summer can have other negative consequences, as climate change is expected to intensify dry-wet cycles, potentially leading to soil degradation (Stirling et al., 2021; Azizi et al., 2019).

## 660   Appendices

### A1. Time series models

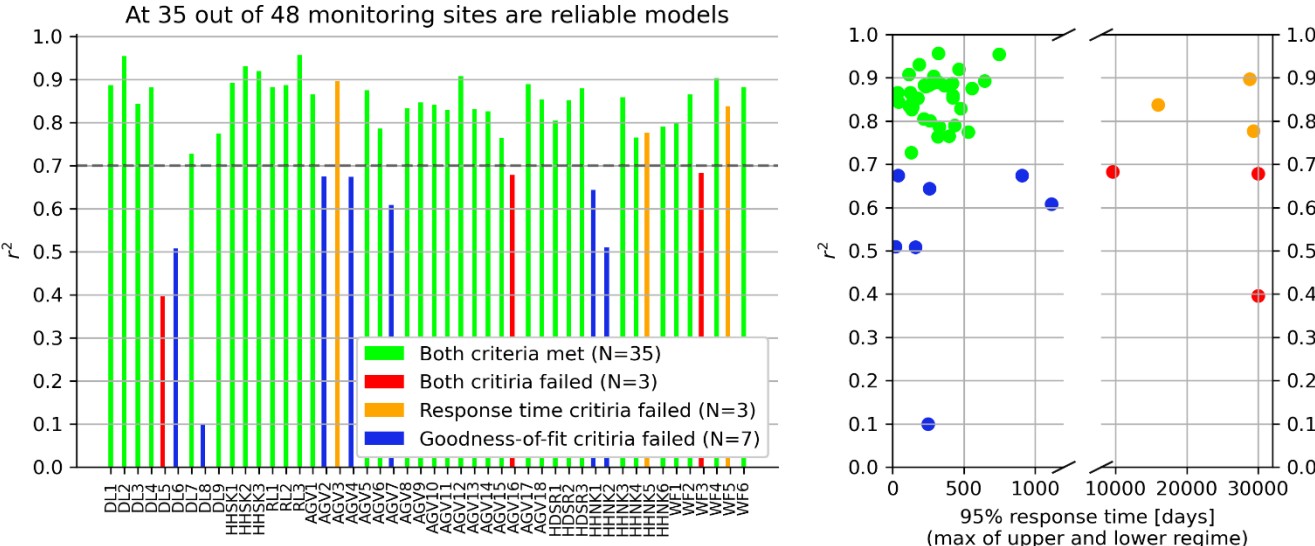

**Figure A1.** *r²* **values of the best-performing TARSO models at each monitoring site, with colors indicating whether the models meet the reliability criteria. Right: Corresponding response times (maximum for both the upper and lower regimes) associated**
**with the R² values.**

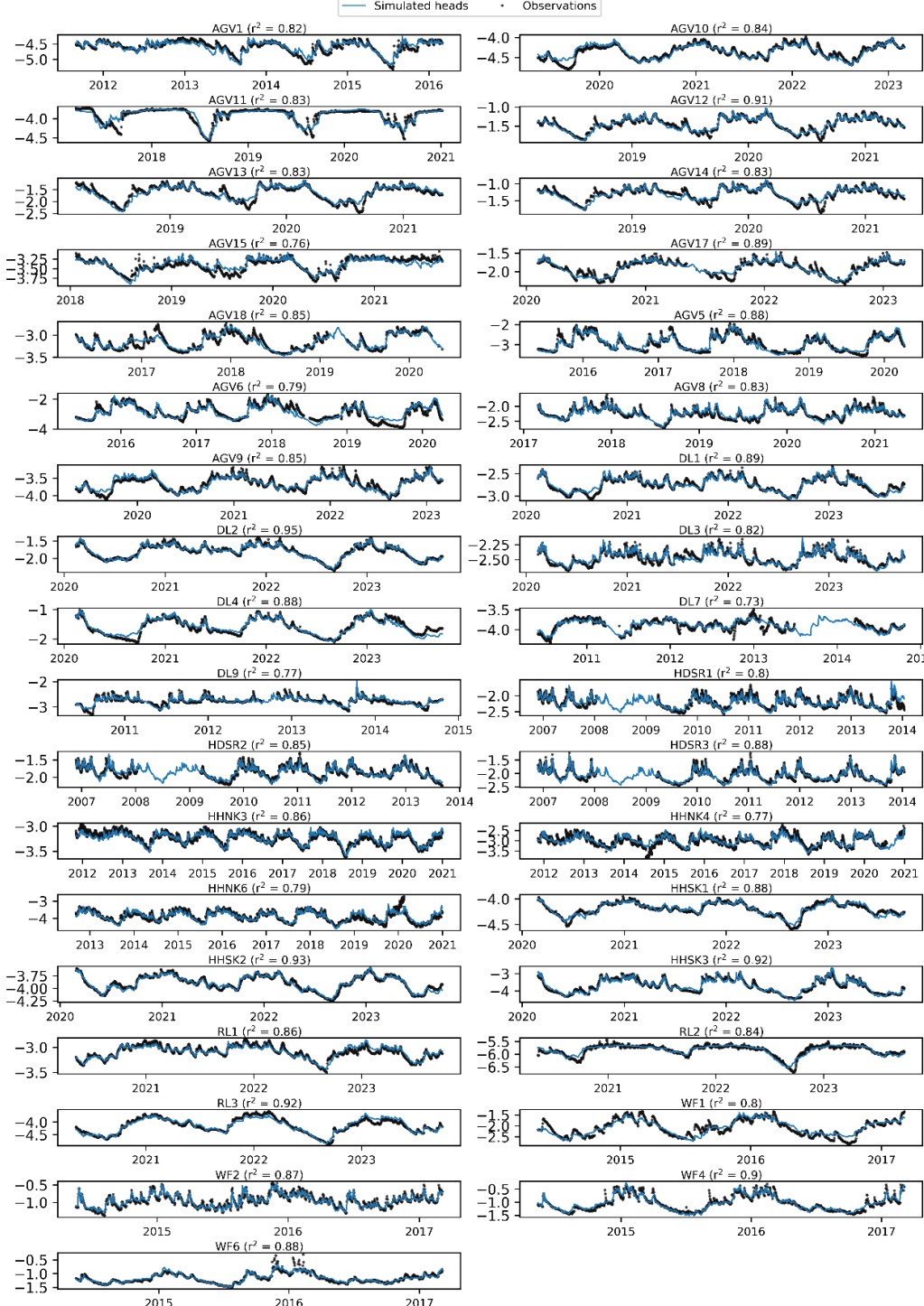

**Figure A2.** The observed and simulated heads for all monitoring wells.

## A2. Seasonality

The dynamics of the peak heads were analysed by quantifying the seasonality of the dikes, which is measured by using the average timing and temporal concentration of the selected head peaks. The method to determine the average timing of head peaks involves circular statistics, and it is extensively described in Hall & Blosch (2018). The average timing of the head peaks is the average date on which peaks have occurred during the time series. The average timing of head peaks can be the result of a wide range of peak dates during the year and therefore the temporal concentration of peaks occurrence within the

year is considered using the concentration index. The concentration index of peak dates around the average timing serves as a measure of how well the seasonality is defined for a specific dike, with 0 indicating evenly distributed peaks during the year and 1 indicating that all peaks occur on the same date.

Seasonality varies across the dikes, but the average timing of these peaks occurs in the winter half-year, as shown in the right

graph in Figure A2. The average timing shifts further into the winter from cluster A to Cluster D, with increasing temporal concentrations. This behaviour is also illustrated by the probability density distributions of the peaks of four dikes in the left graph in Figure A2. The vertical dashed line indicates the average timing, which moves further into the winter, while also the density functions become narrower.

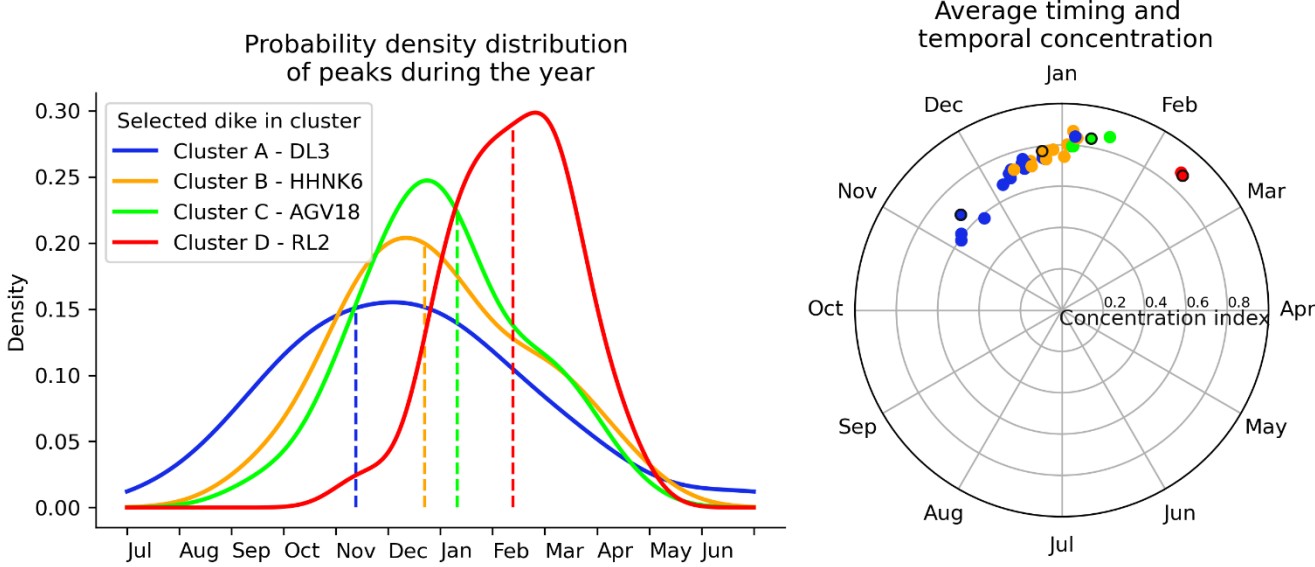


**Figure A3. Right: The average timing and temporal concentrations for the considered dikes, with colors representing the dike clusters (refer to the legend in the left graph). Left: The probability density distribution of the peaks throughout the year for four selected dikes from different clusters, with the vertical dashed line indicating the average timing. These four dikes are highlighted with a black edge around the circle in the right graph.**

**A3. Spatial patterns in dike clusters**

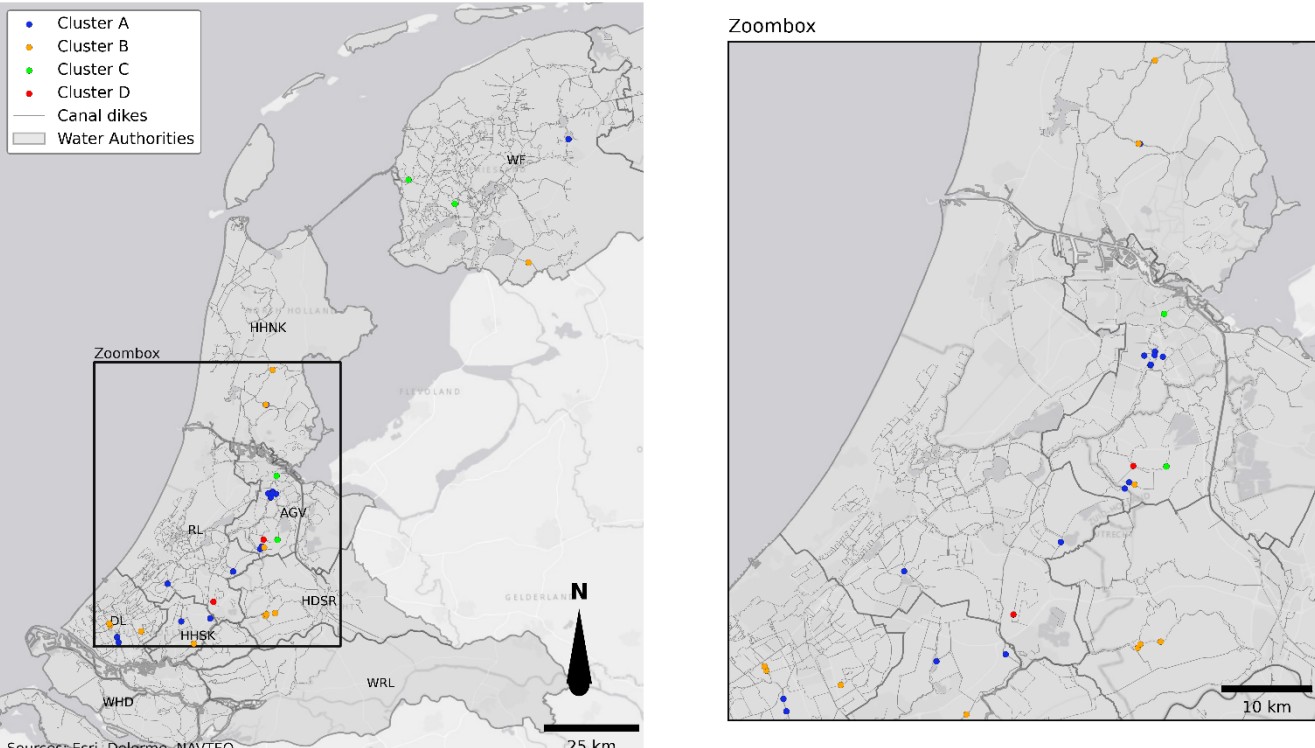

**Figure A4. Mapping of clustered canal dikes across the Netherlands. The left panel shows the full study area, with dikes grouped into four clusters (A–D) based on their peak head response to weather events. The right panel provides a zoomed-in view of the highlighted region, offering a more detailed look at the cluster distribution.**


**A4. Physical dike characteristics and their relation to head responses**

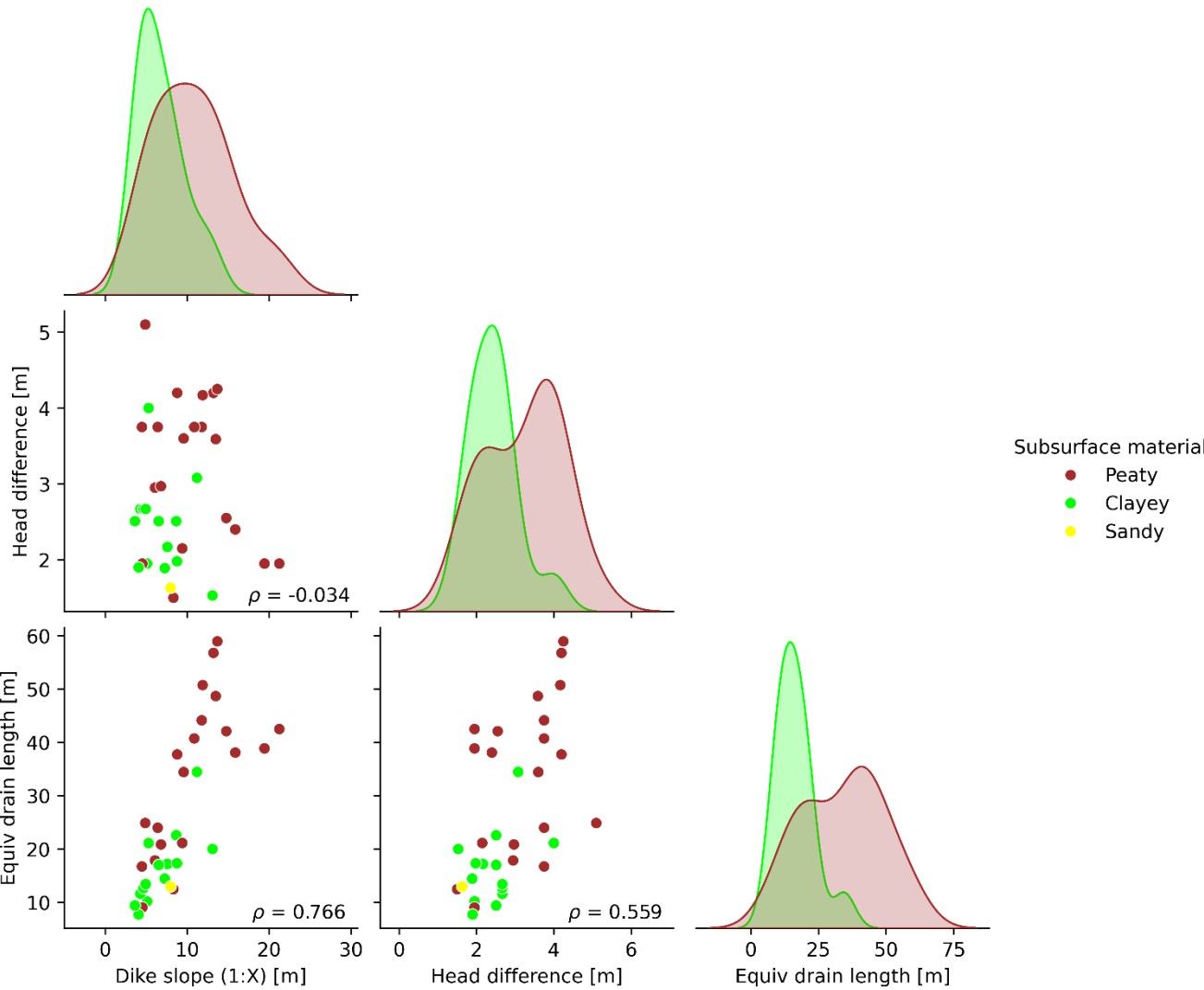

**Figure A5. The relationships between the considered physical dike characteristics are illustrated by scatterplots, with the subsurface material of the dike indicated by color. The Pearson correlation coefficients between the variables are displayed in the bottom right corner. The diagonal plots show the univariate distributions, highlighting the marginal distribution of each variable, with distinctions made based on soil type.**

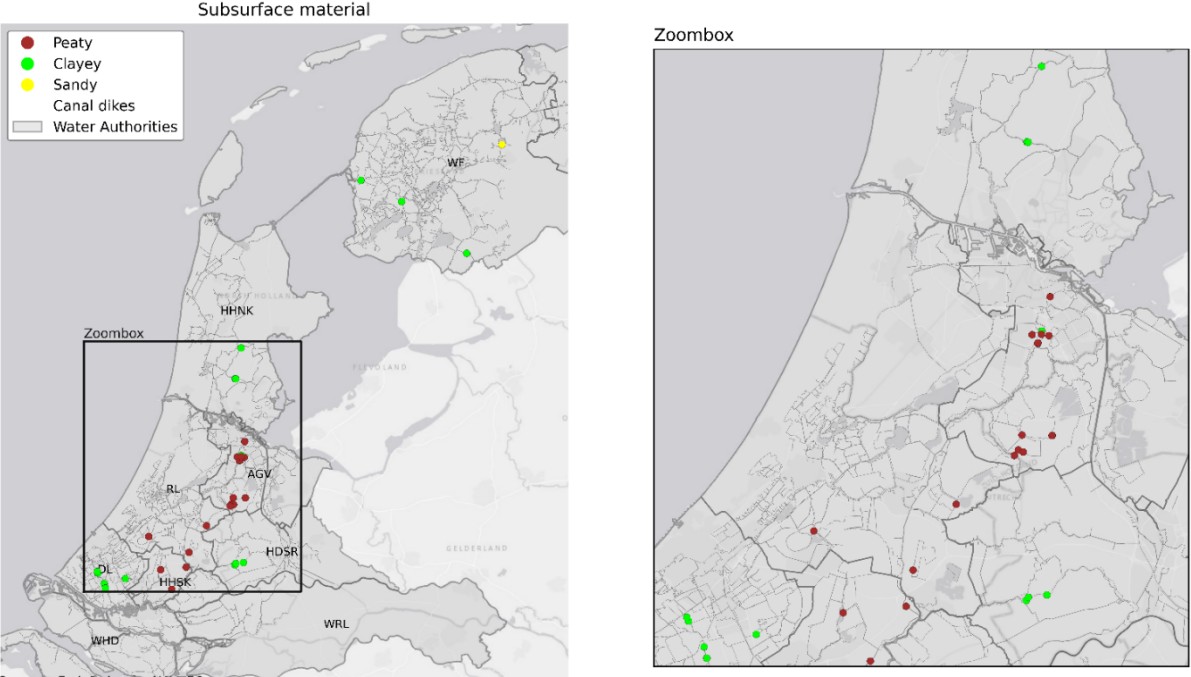


**Figure A6. Mapping of subsurface material (peat, clay and sand) in the 35 canal dikes for which reliable models were developed across the Netherlands.**

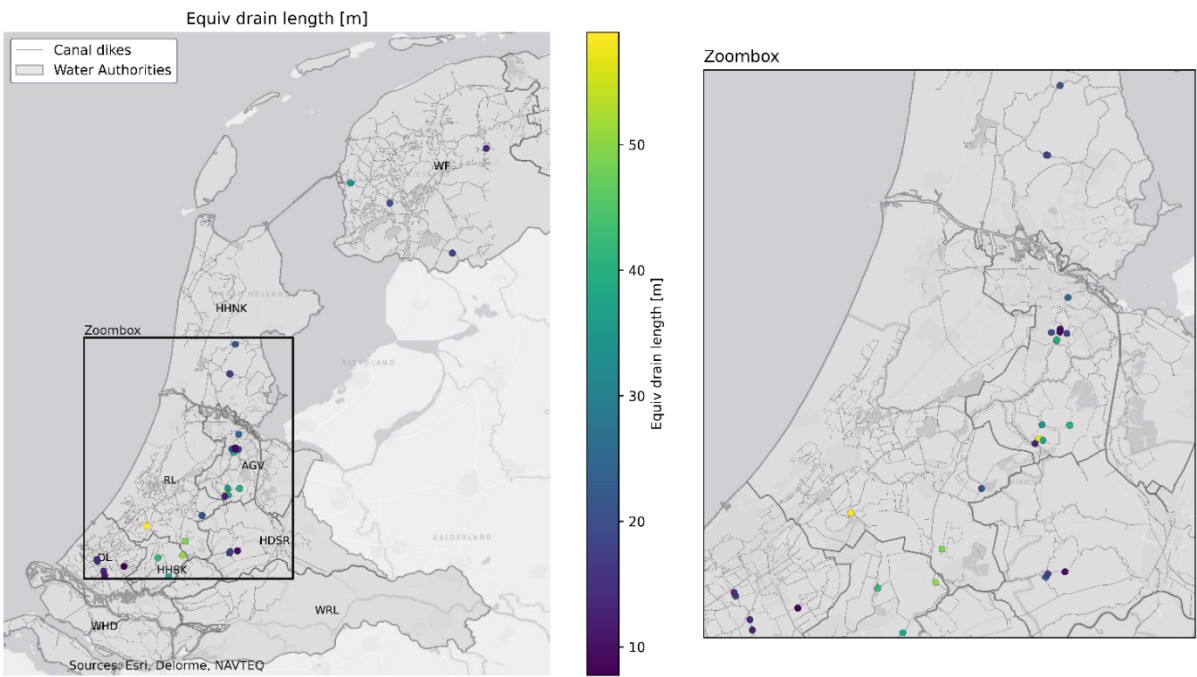


**Figure A7. Mapping of the equivalent drainage length in the 35 canal dikes for which reliable models were developed across the Netherlands.**

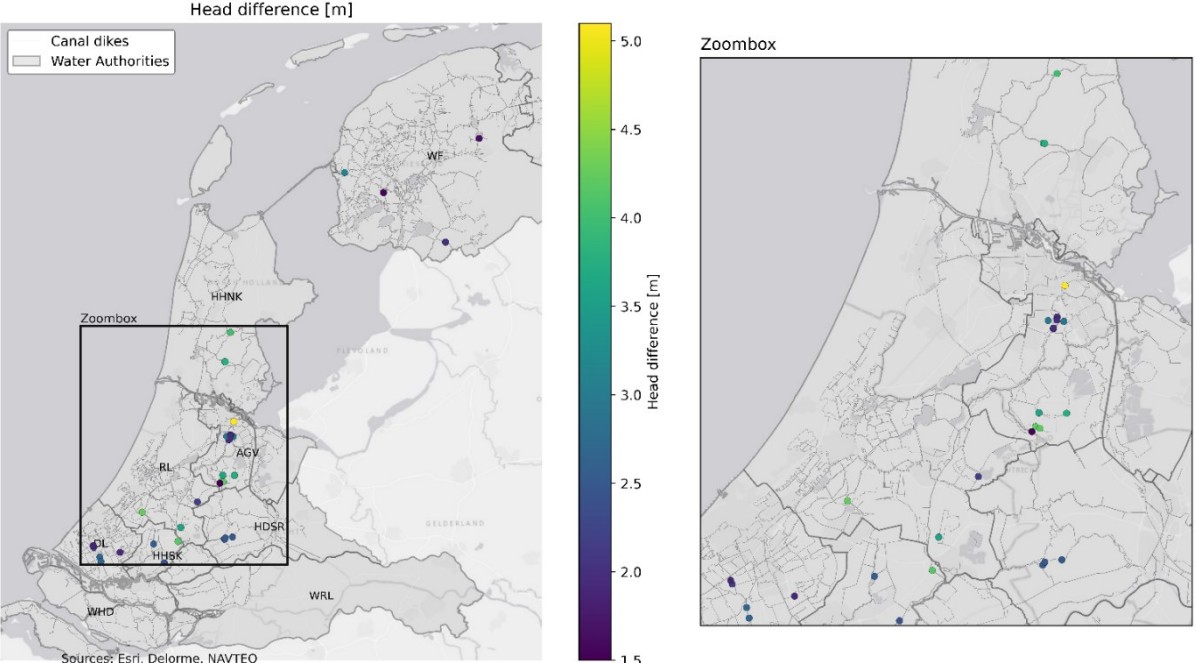

**Figure A8. Mapping of the head differences in the 35 canal dikes for which reliable models were developed across the Netherlands.**


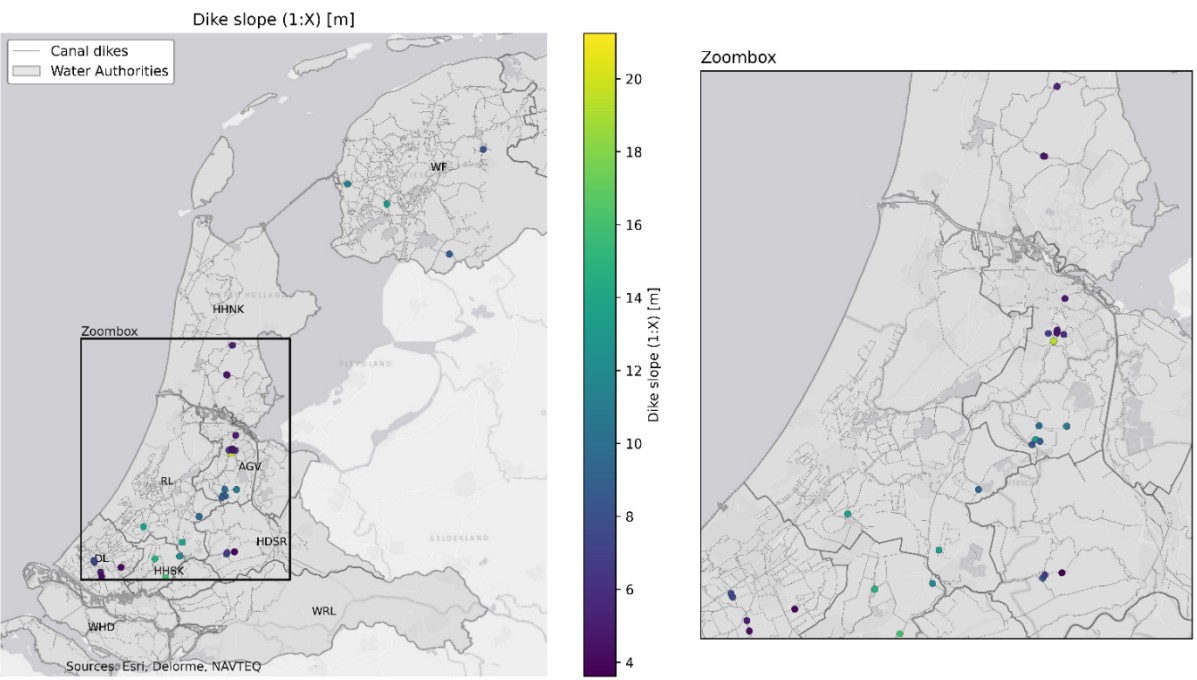

**Figure A9. Mapping of the dike slopes in the 35 canal dikes for which reliable models were developed across the Netherlands.**


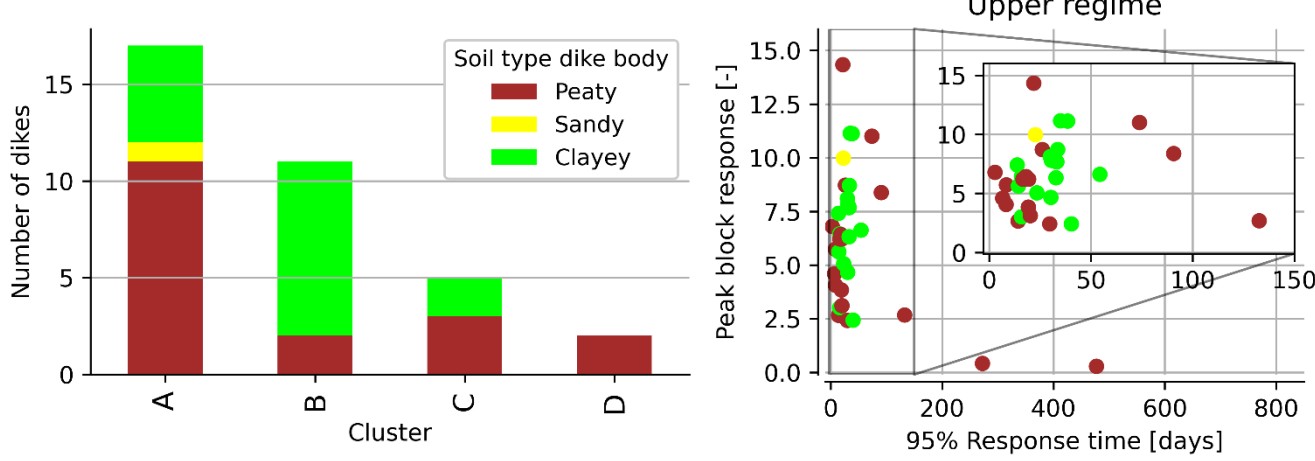

**Figure A10. Stacked bars of the subsurface material of the dike body for the three clusters of dikes. Right: the characteristics of the impulse response functions (95% response time and peak block response) where the colors indicate the subsurface material**

 **A5. Spatial variation in decimate heights**

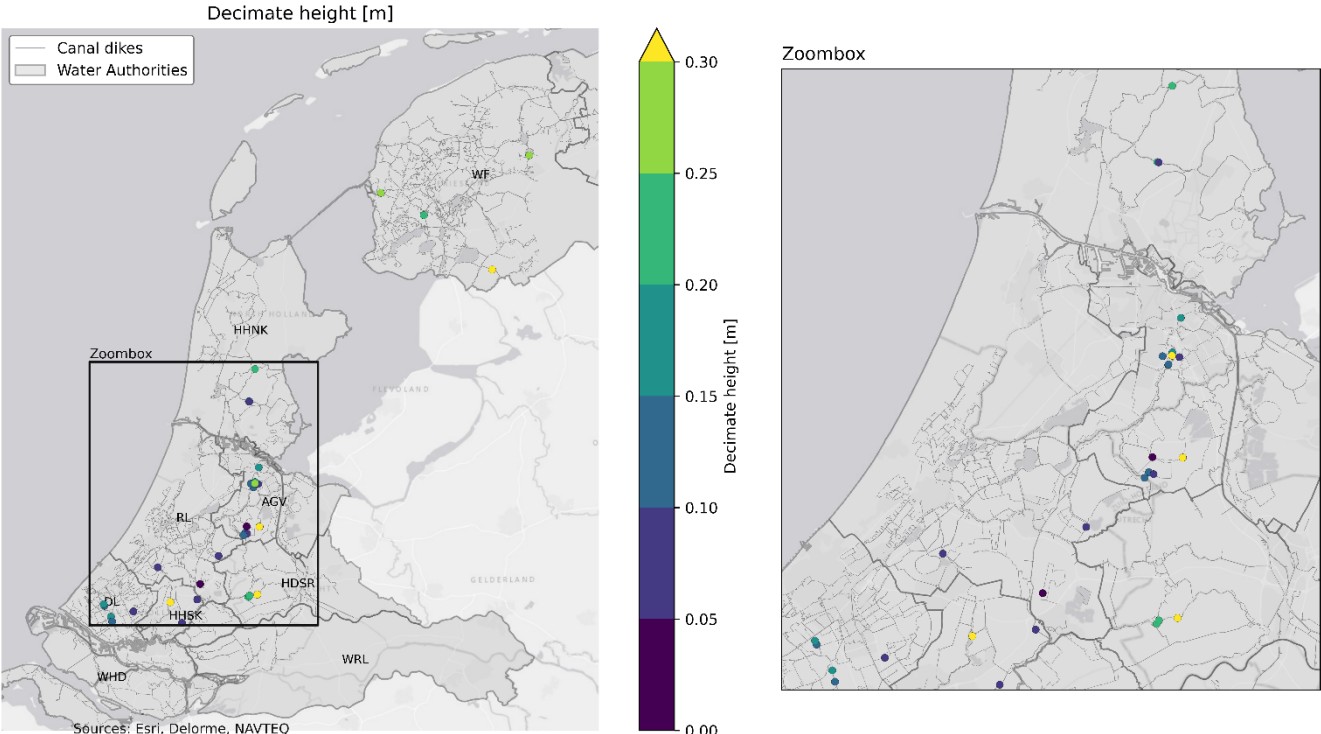

**Figure A11. Spatial variation of decimate height across canal dikes in the Netherlands. The left panel shows the full study area, with different water authorities labelled. The right panel provides a zoomed-in view of the highlighted region. The color scale represents decimate height values. No clear spatial pattern is observed across different regions.**

**Code and data availability statement**

The measurement data used to establish the model set, consisting of 108 head time series across 48 monitoring sites and the local historic rainfall and potential evaporation, are available at 4TU.ResearchData (Strijker, 2024). This dataset contains:

- An overview of the monitoring sites and piezometers, along with their geographic locations (CSV-file and shapefile)
- Hydraulic head time series from the piezometers (CSV-file)
- Time series of local precipitation and potential evaporation (CSV-file)

Furthermore, the script, output data (models and figures) and other relevant data (e.g. dike characteristics and meteorological time-series data representing future climate scenarios) are shared to enhance the reproducibility of this study.

A readme file is added that describes the files in the dataset. The data source has a CC0 license, which entails the waiver of all copyright and related rights, enabling unrestricted use of the data for any purpose. Authors appreciate being informed when using the data by contacting the corresponding author.

**Acknowledgements**

This research was supported by the STOWA and Rijkswaterstaat. Furthermore, the authors express their gratitude to all the water authorities who provided the necessary data: Hoogheemraadschap Schieland & de Krimpenerwaard (HHSK), Hoogheemraadschap Delfland (DL), Hoogheemraadschap Rijnland (RL), Hoogheemraadschap Hollands Noorderkwartier (HHNK), Weterksip Fryslân (WF), Waterschap Amstel, Gooi and Vecht (AGV) and Hoogheemraadschap De Stichtse Rijnlanden (HDSR).

**Competing interests**

The contact author has declared that none of the authors has any competing interests

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
