# Peer review of "The dynamics of peak head responses at Dutch canal dikes and the impact of climate change"

_EGUsphere, 2024_

## Author Comment (AC1)

**Response to reviewers' comments**

"The dynamics of peak head responses at Dutch canal dikes and the impact of climate change" https://doi.org/10.5194/egusphere-2024-1495 submitted to Natural Hazards and Earth System Sciences

We thank the reviewer for her/his thorough, insightful and valuable feedback, both on a general and more detailed level.

Below, we reply to the reviewer's comments and explain how we will address them. The reviewer's comments are shown in *Italicized text in gray*, our responses are shown in blue. We provide detailed responses to the major comments, along with specific **actions** to improve the manuscript. For minor comments, we offer brief responses, as we will incorporate these suggestions to enhance clarity and refine terminology throughout the text.
* * *
**Anonymous Referee #1**

*The paper reports on a comprehensive analysis of dike strength and failure for a large number of polder dikes across the Netherlands. Using a model for dike 'ground'water head calculation and time series of precipitation and evapotranspiration, the model produces time series of water heads in dikes. Results are statistically analyzed, including grouping of dike types across the area.*

*The subject fits well to the scope of NHESS, and addresses a field that has received increasing attention, both in water management and (applied) research into dike stability and flood risk.*

Thank you for your positive feedback; we appreciate the recognition of our study's relevance to dike stability and flood risk research.

*Still the manuscript requires revision before final publication. My main points are:*

- *I miss a spatial component in the discussions of clusters and 'correlation': how would a map indicating cluster member of dike sections look - is there any spatial correlation, or not?*

  We acknowledge that the spatial component of clustering and correlation is not explicitly discussed in the paper. However, the coincidence matrix in Figure 9 provides an initial indication, showing that each cluster includes dikes from multiple water authority regions (different IDs). Additionally, Figure 8 suggests no clear spatial correlation in the impulse response characteristics, as noted in Lines 367–368: "*Initially, the variation of head responses across regions suggests no specific pattern, likely due to the heterogeneous subsoil conditions within the canal dike system, as shown by the random color distribution in Fig. 8.*"
  Nevertheless, we acknowledge the lack of a spatial discussion of the clusters and a map visualizing cluster members, which would add relevant insights.

  **Action:** In the revised manuscript, we will include a spatial analysis and maps illustrating:
  - The spatial distribution of clusters, and
  - The sensitivity of dikes based on head decimate heights.

  These findings will be integrated into the relevant sections—Section 4.2.1 (clusters) and Section 4.3.1 (decimate height sensitivity). To maintain readability, the maps will be placed in the appendix.

- *The meaning of the clusters determined from the statistical analyses remains a unclear - what do they tell us, and how do these contribute to the overall objective, i.e. predicting dike stability over large areas, under conditions of extreme precipitation. I got the impression that a different type of clustering would have added more insight.*

The clusters in this study group dikes based on similarities in head peak responses to the same weather events. This was determined by analyzing the coincidence of simulated head peaks over 30 years of rainfall and evaporation (Lines 388–389). These clusters represent variations in head response across the dike system, which is relevant for regional risk assessments. Since extreme loads may result from different weather events in different clusters, this variation affects system reliability and has direct implications for the flood risks in polders, as discussed in the introduction (Lines 56–57) and Section 5.2 (Lines 532–544). Additionally, these clusters with different head responses can help identify potential instabilities based on predicted rainfall. Therefore, it would be even more useful to understand how dikes with specific characteristics respond to heavy rainfall or which types of dikes exhibit certain head response patterns. Both aspects are examined in Section 4.2.2, where we analyzed potential relationships between physical dike characteristics, dike clusters, and impulse response function parameters. The lack of clear relationships suggests that general attributes such as soil type or geometry do not directly determine head response behaviour. This is an important finding, as it indicates that commonly used physical characteristics cannot be linked to the head response and it is likely influenced by more local dike properties, for example local subsurface . This is also clearly stated in the conclusions.

Regarding the reviewer's comment about different type of clustering, we are unsure what specific approach is being suggested. However, we believe this may relate to the detailed comment on Lines 586–587, which we address separately further below.

**Action:** We will revise Section 4.2 to clarify the meaning of the clusters and how they contribute to the overall objective of predicting dike stability under extreme precipitation. Additionally, we will refine related discussions in other sections based on the reviewer's detailed comments.

- *The discussion section should me restructured and complemented with thoughts about climate change impacts, and implications for the dikes, and their management.*

This study quantifies the impact of intensified rainfall and drier summer periods—one of the key impacts of climate change—on peak head levels, one of the key elements for dike stability. However, it assumes that the head response remains unchanged over time. The potential impact of non-stationary responses, such as variations in hydraulic conductivity due to repeated dry-wet cycles, is not explicitly addressed, which may be what the reviewer is referring to. Other climate change impacts on dike stability and management, like deterioration processes and other failure mechanisms e.g. horizontal translation, are also relevant but are not currently discussed in detail. While Section 5.2 already addresses some management implications, we recognize that the reviewer is requesting a more comprehensive discussion.

**Action**: We will restructure the discussion section to improve clarity and explicitly include additional climate change impacts. Furthermore, we will expand on the implications for dike safety and management to provide a more detailed perspective.

- *The conclusions are too much a summary: rewrite these in 'conclusive' style, not a story of what you did.*

  We acknowledge the reviewer's feedback regarding the conclusion section. While we aimed to summarize the key findings, we understand the need for a more conclusive style that emphasizes the main insights rather than a summary of actions.

  **Action:** We will revise the conclusions to present the findings in a clearer and more conclusive style, focusing on their significance rather than repeating the steps taken in the study.

- *At several points the text is wordy, or unclear - see specific comments.*

  We appreciate the reviewer's feedback on clarity and conciseness.

  **Action**: We will critically review the manuscript, using the specific comments as guidance, to improve clarity and make the text more concise. Our revisions will focus on eliminating unnecessary wording while ensuring that the key messages remain clear.

*In the attached pdf I have put my detailed comments.*

We appreciate the detailed comments, of which many are focusing on terminology and clarity. The comments are incorporated in the revised manuscript to make the text more readable. We would like to highlight a few detailed comments and give a reaction:

- L586-587: *"so, this makes me doubting whether your clustering resulted in something that contributed to reaching your objective, instead of a statistical of sections by means of numerous characteristics as such."*
  We have no doubts about the validity of the clustering approach and its contribution to the study's objectives. Our goal is to assess the dynamics of peak hydraulic heads in canal dikes at a national scale, specifically in response to heavy rainfall events, by analyzing variations in head responses and head statistics. The clustering was designed to identify groups of dikes with similar head response behavior, making it a valuable tool for regional risk assessments. This approach helps to understand the variation in loading conditions and distinguish which dikes may experience similar loading conditions under extreme weather events (see also our response in the major revision). Therefore, we examined the relationship between dike characteristics and head response in Section 4.2.2 to determine whether physical attributes could explain (1) the clustering and (2) the impulse response function parameters. While no clear relationships were found, this is an interesting finding in itself, as it suggests that general dike characteristics alone may not be sufficient to predict head response behavior.

- L416-*423:* *"part if this is method, furthermore, I prefer reading about what causes behaviour, insead of listing all sorts of tests and p values - these distract here; instead describe which factors show (functions / causal) relationshops (underpin that with test results), that is interesting to the reader"*
  We appreciate this suggestion and understand the preference for focusing on causal relationships rather than listing statistical tests and p-values. While these tests are important

for validating our findings, we agree that emphasizing the key factors driving head response behavior will improve readability and clarity.

**Action**: We will revise this section to shift the focus toward explaining which factors show functional or causal relationships with head response behavior. The statistical results will be used to support these insights rather than being presented as a list. We will move the description of various tests to the methodology section, where we make a new subsection named "3.3. Statistical tests for relationships".

- Line 449: *"it would be interesting to know how much height variation is still acceptable and 'safe' is that 10 cm, 50 cm, or 2 cm?"*
This is indeed an interesting and important question. However, determining the exact height variation that remains acceptable and "safe" is complex and depends on many factor regarding the strength of dikes. While this falls outside the scope of the current study, it is a key focus of our follow-up research!

---

## Author Comment (AC2)

**Response to reviewers' comments**

"The dynamics of peak head responses at Dutch canal dikes and the impact of climate change" https://doi.org/10.5194/egusphere-2024-1495 submitted to Natural Hazards and Earth System Sciences

We thank the reviewer for her/his thorough, insightful and valuable feedback, both on a general and more detailed level.

Below, we reply to the reviewer's comments and explain how we will address them. The reviewer's comments are shown in *Italicized text in gray*, our responses are shown in blue. We provide detailed responses to the major comments, along with specific **actions** to improve the manuscript. For minor comments, we offer brief responses, as we will incorporate these suggestions to enhance clarity and refine terminology throughout the text.
* * *
 **Anonymous Referee #2**

*This manuscript outlines an application of time series models using impulse response functions to model the hydraulic heads observed with dike systems in the Netherlands. Different model structures are tested to simulate the heads, with a nonlinear-threshold model (TARSO) found to perform the best. This is an interesting result, that could teach us something about how the heads in dike systems respond to precipitation and potential evaporation. The study attempts to relate model characteristics to various physical characteristics of the dike systems, with moderate success. I generally found the manuscript well written and the figure quality appropriate. The topic fits the scope of the journal. I have a couple of major comments that should be addressed, and some minor technical comments at the bottom.*

Thank you for your constructive feedback; we appreciate your insights and will carefully address your comments.

- *One thing I was missing in the manuscript is an explanation and interpretation of why the threshold nonlinear model structure (TARSO) works best for the hydraulic heads in Dikes in the Netherlands. This is a surprising outcome to me, that deserves more thought and might be informative for future attempts to model the heads in dikes. This model was designed for a different type of system (groundwater levels in polders, influenced by ditches falling dry and being activated). Perhaps there is topping-off of the heads in dikes. These types of models are commonly used to gain understanding of how groundwater systems function, and why. A discussion of this type is currently missing from the manuscript but would be a welcome addition.*

  The strong performance of the TARSO model in modeling hydraulic heads in Dutch canal dikes can be explained by the non-linear characteristics of the head response in these dikes. The manuscript gives some suggestions in line 330 – 332: "*This non-linear behaviour can be the result of various soil layers in the dike body, each with distinct hydraulic properties, and changes in infiltration rates or nonconstant storage capacities of the unsaturated zone during the dry season*" The suggestion that there may be a "topping-off" effect in dike heads is an interesting perspective that aligns with the need to account for non-linear responses.

**Action:** We will expand on this discussion in the revised manuscript, providing a more extensive interpretation of why the TARSO model performs well for dikes. However, as these explanations remain hypotheses at this stage, we will also emphasize that further research is needed to verify these mechanisms.

- *Looking at Figure 6, I was very surprised by the simulated behavior of the FlexModel given that all models share the same input and not be that far off from each other. I made a quick script to model the data using the Pastas default values to better understand the result but got an average of R2=0.68 for the FlexModel, much higher than the reported value of 0.32. I suspect some suboptimal choices were made for that model. Perhaps the Authors can revisit the scripts and double-check this, or explain this result in more detail.*

Thank you for this remark. It is unclear how the reviewer obtained an average R² of 0.68 for the FlexModel. When we apply the FlexModel using default values, we obtain a significantly lower average R². We have experimented with different initial parameter settings but were unable to achieve a substantial improvement in model performance.

If the reviewer is willing to share their script, this would be very helpful for comparison and to better understand the differences in our findings. Additionally, this relates to the subsequent comment regarding the reproducibility of results, which we address separately.

- *I assessed the manuscript for its reproducibility. I appreciate the authors providing the original head, precipitation, and evaporation data is provided. This data provides a unique dataset on head measurement in dike systems, which might be worth highlighting in the manuscript. I note here that none of the data underlying the results shown in the figures and tables are shared, nor are the scripts that lead to the results. This makes it difficult to verify the results and/or build upon this work. I would recommend the authors to share the scripts and output data on a FAIR repository to improve the reproducibility of this study.*

Thank you for assessing the reproducibility of our study and for highlighting the uniqueness of the dataset. We appreciate the importance of making research more transparent and reproducible.

**Action**: We will emphasize the uniqueness of the dataset in the manuscript. Additionally, we will share the scripts and output data on 4TU.ResearchData, ensuring alignment with FAIR principles to improve the reproducibility and accessibility of our work.

*Minor technical suggestions:*

*L84: Potential evaporation* Correct; will be revised

*L195: Figure 4 and 5 appear to be the same. I am not sure if another figure is meant to be here. Otherwise Figure 5 can be removed.* Correct; we will remove Fig. 5.

*L192: How well does GeoTop actually work for dikes? I can imagine that these are not related at all, given that dikes are built by humans with specific materials. This may influence the results later on relating outcomes to the soil types, i.e., would the relationship with the soil type improve if the GeoTop data is left out? Some consideration about this would be good here.*

This is a good suggestion; We will check this and write down the findings in the revised paper.

*L238: I think Sm is substituted by R, not the other way around.* Correct; will be revised

*L276: simulate "the heads".* Good suggestion, will be revised

*L288: How was this threshold of 0.7 determined? How sensitive are the result to changing this threshold.* We will elaborate on the sensitivity of this threshold.

*L302: Analyses* Will be revised

*L303: every "…"?* We will rewrite this sentence: "of every" will be deleted.

*L327: r2 was previously referred to as R2, check throughout* Will be checked and corrected in the manuscript

*L329: The FlexModel is a nonlinear recharge model, the other three models are not. I think the TARSO model is meant here, which still computes recharge using a linear equation.*

In literature, tha TARSO-model is often referred to as a nonlinear model, since it accounts to some extent for the nonconstant relationship between precipitation excess and water table depth caused, in contrast to linear model (Knotters en Gooijer, 1999)

*L342: replace by "in the summer of 2019"* Will be revised

*L343: disturbances* Will be revised

*L345: scatter plot in Figure XX.* Sentence will be rewritten.

*L366: I don't understand what "peak block" is, please clarify.* This is explained in lines 347-349.

*L496: Apparently, the head time series were filtered using some reliability criteria in this study. This should be mentioned in the section describing the data. What reliability criteria were used?*

This is discussed in section 3.1.3 – Model calibration and selection. Lines 287-294 describes the reliability criteria used

*L501: Remove "explicitly". Uncertainty was not considered, as I understood from the manuscript.*

We believe you mean line 511: we will remove "explicitly".

Knotters, M., & De Gooijer, J. G. (1999). TARSO modeling of water table depths. Water Resources Research, 35(3), 695-705.

---

## Author Response (AR2)

**Response to reviewers' comments – Minor revisions**

"The dynamics of peak head responses at Dutch canal dikes and the impact of climate change" https://doi.org/10.5194/egusphere-2024-1495 submitted to Natural Hazards and Earth System Sciences

We thank the reviewers for their valuable feedback, both on a general and more detailed level.

Below, we reply to the reviewer's comments and explain how we will address them. The reviewer's comments are shown in *Italicized text in gray*, our responses are shown in blue. We provide detailed responses to the major comments, along with specific **actions** to improve the manuscript. For minor comments, we offer brief responses, as we will incorporate these suggestions to enhance clarity and refine terminology throughout the text.
* * *
**Anonymous Referee #1**

This reviewer had no further suggestions for revisions and accepted the manuscript as it is for final publication.
* * *
**Anonymous Referee #2**

*I re-reviewed this manuscript after the first revision round. The comments from the previous round were adequately addressed, and the revision substantially improved the manuscript in the presentation and discussion of the results.*

Thank you, and we are glad to hear that the manuscript has been improved.

*Only one important issue remains with the application of the nonlinear recharge model, that should be addressed before publication. I could identify the issue after the scripts were published. The error is that the input data for the model was provided in meters/day, whereas the model requires mm/d. Changing this will significantly improve the model fit and alter the results, and possibly the conclusions.*

*After rerunning the scripts and updating the description/figures of the results, I think this manuscript is ready for publication.*

We have changed the units, and the resulting Flex models have improved significantly. We have changed the script at the digital repository, including Figure 5 in the manuscript.

Furthermore, the metrics in the first paragraph of section 4.1.1 have been updated. Still, the TARSO model performed overall the best. Therefore, this will not affect the conclusions of the paper.
* * *
**Anonymous Referee #3**

*This study presents a compilation of hydraulic head measurements from polder regions across the Netherlands and conducts a data-driven time series analysis of current and future trends in head. The authors find that a non-linear threshold model best reproduces the observed data, and observation*

*data may be clustered into 4 groups based on coincident observations of peak head in space and time. Under four different climate scenarios, changes to the return interval of extreme head values may be negative or positive because future outcomes depend on the change in future hydroclimatic variables. This study is likely to be of interest to the scientific community, particularly to those involved in hazard assessments of low-lying canal dikes. I look forward to seeing it in print.*

Thank you for your positive feedback; we appreciate the recognition of our study's relevance to dike stability and flood risk research.

*The manuscript would benefit from a round of minor revisions. Suggested edits to improve the manuscript primarily relate to manuscript organization. There are multiple places where content in the Introduction, Methods, and Results would fit better in a different section. In addition, the results of the climate change analysis are interesting and will be of interest to the community, but they merit further attention in the Discussion section. Lastly, there are some grammatical errors to resolve. Specific recommendations on each of these themes (as well as a few minor ones) are presented by line number below.*

- *Line 12-13: I recommend rephrasing this sentence to: "These models were used to simulate 30-year time series of head under current and plausible future climate scenarios."*
Rephrased: "… under current and future climate scenarios"

- *Line 56: remove comma after "heavy rainfall event"* Removed

- *Lines 84-98, and Figure 2: This paragraph and Figure 2 are useful to understanding the methodological framework of the study, but they feel out of place in their current location at the end of the Introduction section. This paragraph and figure summarize the Methods used in the study; for this reason, I suggest moving both the paragraph and Figure 2 to the very start of the 3. Method(s) section, before section 3.1. This will fit more naturally within the structure of the manuscript and will give readers a helpful summarization of methodology right before diving into all of the details. Note that this change will also require renumbering Figures 3 & 4 and their respective references in the text. The final sentences of the previous paragraph (i.e., lines 78-82, starting with "This study aims to…") seem like a more natural end to the Introduction section, as they provide a general roadmap of the study. The authors may also choose to expand on the roadmap with an additional statement or two on the novelty of their approach and the research gap that they address.*
We have moved lines 84–98, along with Figure 2, to the beginning of Section 3.1 to better align with the manuscript's structure, as you recommended. We also added a brief statement highlighting the novelty of our approach at that point in the text. The research gap remains addressed in the Introduction.

- *Line 85: the acronym "IRF" is defined in three separate places in the text (here, line 193, and the caption of Table 1), but "IRF" is only used as a column header in Table 1. I would recommend removing this acronym from the text since it is not used, or alternatively, replacing the following instances of "impulse response function" in the text with "IRF".*
We have removed the acronym IRF and replaced the header in Table 1

- *Line 89: remove comma after "resulting in a set of models"* Removed

- *Line 110: fix spelling of "focusses"* Adjusted

- *Figure 4: I find this figure to be very helpful. One recommended change: it is not clear on first glance what is meant by the terms "90th percentile profile" and "10th percentile profile" – my first interpretation was that they referred to the phreatic surface profiles. These terms are eventually defined as points drawn from elevation profiles on line 309. Please make this clearer in either the figure itself or in the figure caption.*
  We have added a sentence to the figure caption: "The 90th and 10th percentile profile points were derived from the elevation profile as an approximation of the dike slope."

- *Line 197: the index of summation under the summation symbol is a capital "M", the same as the upper bound of the summation. Should this be a lower-case "m"?* Yes, lower-case is changed to "m"

- *Line 205: I understand that the authors use more than one type of response function (e.g. exponential, gamma) that are not defined in the manuscript. This seems fine, but can the authors provide a citation here to point interested readers to a resource with fuller descriptions of the response functions?* We have added a reference to line 205.

- *Line 215: this water budget does not include surface runoff (follow-up: as described in the Discussion section 5.1); is surface runoff unimportant/less relevant to polder canal hydrology? This merits a brief mention here, too.*
  We acknowledge that surface runoff may be relevant for canal dikes, and we have clarified this in the text. In the following paragraph, we also elaborate on nonlinear recharge models, such as the FlexModel, which are capable of accounting for processes like surface runoff. A sentence has been added to make this distinction clearer to the reader.

- *Table 1: please replace all instances of "lineair" with "linear"* Replaced

- *Lines 246-249 and Figure 5: These results belong in the Results section.* We have incorporated these results in section 4.1.1.

- *Line 296-298: though not entirely clear, this sentence seems to imply that only the exponential distribution was used throughout the study. If this was the case, then for clarity, I would recommend removing the discussion and definition of the general form of the generalized Pareto distribution and instead define just the exponential distribution. If not, then please clarify where/how the Type I, II, and III distributions were utilized in the text.*
  The generalized Pareto distributions were used for a sensitivity analysis to explore the impact of different tail behaviors on the results. To clarify this, we have added a sentence to the text explaining the purpose of including the full distribution family, while noting that the exponential distribution was used in the main analysis.

- *Line 310: as with "IRF", the target levels acronym "TL" is not used in the text (other than in Figure 4) and should be removed. Its definition in the caption of Figure 4 should remain.*
  Removed

- *Line 317: "clusters of dikes" is mentioned here, but no methodological details are provided. They are instead described later in the Results section 4.2.1 (lines 394-411). Most of the paragraph on lines 394-411 belongs in the Methods, not the Results.*

We have moved lines 394-411 from the Results to the Methods section and added a subsection named "Clustering head responses".

- *Section 4. Results: there are several places where interpretation of results (and attribution of causes behind the results) are included in the Results section but likely belong in the Discussion section. Moving them would help focus the reader's attention on the results of this study while helping to bolster the discussion of study results within a wider scientific context, which is largely missing from the Discussion section. In general, I encourage the authors to more fully differentiate presentation of results in the Results section from interpretation of results and their connection to past work in the Discussion. The lines below are suggested places to consider moving content from the Results to the Discussion:*

This is a valuable comment. We agree that interpretation within a broader scientific context belongs in the Discussion section, and we have moved relevant content accordingly. At the same time, we note that some degree of immediate interpretation—such as explaining observed differences or trends—can help readers better understand the results. It gives a better narrative flow and practical understanding. Therefore, we prefer to keep brief, descriptive interpretations in the Results section where they enhance understanding, while ensuring that broader analysis and comparisons with previous studies are addressed in the Discussion.

  o *Lines 379-386: beginning with "The difference in response time can be…"* We prefer to keep this text in the Results section. These lines offer explanations for observed differences and are directly tied to specific results, which help the reader understand why certain patterns appear and improve clarity. We have slightly rephrased the text to make it clearer that this is an interpretation of the results.

  o *Lines 473-479: all sentences on these lines* This section has been moved to Discussion Section 5.2, as it is more appropriately considered an implication of the results.

  o *Parts of lines 498-503 that infer explanations for the results* We have chosen to keep this text in place but have rephrased it to clarify that it represents an interpretation of the results. It serves to explain the observed outcomes and supports the reader's understanding of the findings. The subsequent comments and responses provide a more detailed explanation of changes in these lines.

- *Line 330: recommend using "Therefore," in place of "So,"* Replaced

- *Figure 6 and lines 350-351: what is "A" in the block response plot? It seems to be the peak response value. Additionally, lines 350-351 state "The peak of the block response represents the maximum increase in head level that would occur," implying that the block response has dimensions of length (e.g., units of meters), but the y-axis label in the block response plot is "Response [-]", implying it is dimensionless. Can the authors please clarify?*
"A" in the block response plot represents the peak of the block response function. The block response itself is dimensionless, as it describes the unit response of the system to a unit input (i.e., 1 mm of recharge). To clarify this, we have revised the figure caption and clarified this in the main text.

- *Line 425-426: This first sentence is unnecessary; it can be removed without loss of detail.*
Removed

- *Lines 464-466: The definition of decimate height and how it is used would be more appropriate in the Methods section, probably section 3.2.*
We have moved it to Section 3.2 and expanded on it following the description of the extreme value distribution.

- *Line 485: little information is provided in the manuscript on the climate change analysis portion of the study. Greater description of the datasets presented in section 2.2.2 would be helpful to the reader, i.e., fuller descriptions of SSP5-8.5, SSP1-2.6, and the wet- and dry-trending scenarios that were introduced on line 176.*
A more detailed description of the datasets is given at the end of section 2.2.2.

- *Sentences from this section that describe methods should be moved to an appropriate section in the Methods (e.g., lines 485-488, the definition of the probability factor on line 494-495). The authors might consider a new section at the end of the Methods to bring the descriptions of climate change data and methods together in one place.*
We have moved the definition of the probability factor to Section 3.2, where the extreme value analysis is discussed, along with the decimate height. Both the decimate height and the probability factor are metrics used to interpret and compare extreme value distributions. We do not think a new subsection is necessary.

- *Line 502: the authors state that the "precipitation increases" in the climate scenarios with drying trends; this seems like a contradiction, although it may not be. Perhaps additional details on the methods behind the climate-change analysis (as suggested above) would alleviate this confusion.*
We have added a more detailed description in section 2.2.2. about the climate scenarios and we rephrased this paragraph.

- *Lines 503-504: The authors state "For climate scenarios in 2100 with a wetting trend, the frequency of occurrence of extreme load levels is increasing," but the right plot in Figure 11 seems to indicate that the low-emission, wet regional response scenario (Lw) results in little-to-no change in probability factor. Please clarify.*
We have revised the text to provide a more comprehensive clarification of the results, with a clearer distinction made between emission scenarios, regional responses, and time horizons. The sentence noticed by the reviewer is also revised: "*By 2100, however, high-emission scenarios indicate an increase in the frequency of extreme head levels, caused by wetter winters, with the most significant impact observed in the wetting regional response.*"

- *Line 516: change "Frist" to "First"* Changed

- *Lines 629-631: One important aspect of the Conclusions and the Abstract is that extreme peak heads "…are expected to occur between 3 times less and 8 times more frequently by 2100, depending on the climate scenario and the type of canal dike," but this statement is not expounded upon in the Discussion. In addition to the above-mentioned modifications of the Results/Discussion sections, I would recommend a few sentences in the Discussion that explicitly tease this conclusion apart. For instance, under which specific scenarios (and*

*timeframes) do the extreme peak heads tend to become less frequent, and under which do they become more frequent? The right plot in Figure 11 also shows some interesting results that are not discussed: the high emission + drying scenario (Hd) shows large declines or no change to the probability factor in 2050 but generally shows increases in 2100. Likewise, the high emission + wetting scenario (Hw) shows a slight increase in 2050 but a much larger increase across the board in 2100. Can the authors infer any causes behind these temporal trends?*

Again, a valuable comment. We have rephrased Section 4.3.2 to more clearly elaborate on the statement that extreme peak heads are expected to occur between 3 times less and 8 times more frequently by 2100, depending on the climate scenario and the type of canal dike. This revised section now includes a clearer breakdown of the results by emission scenario, time horizon (2050 vs. 2100), and regional response (drying vs. wetting).

In addition, we have expanded the Results section by explaining the differences between scenarios and the reasons behind them, and we now explicitly interpret the temporal trends visible in Figure 11. These additions help clarify how the frequency of extreme peak heads changes across different timeframes and scenarios. Although these interpretations remain within the Results section, instead of the Discussion section, we believe they now provide the detail needed to better support the conclusions and address the reviewer's suggestion.

In the second paragraph of the discussion section, we also elaborate on these findings by comparing them to other literature from landslide studies. This also gives a broader scientific perspective.

- *Finally, a discussion of precipitation seasonality is mentioned in the Abstract, Introduction, Conclusions, and the Appendix, but is almost entirely absent from the Discussion. Trends in seasonality and the importance of the type of canal dike when considering future climate conditions should be explicitly commented upon in the Discussion.*

Changes in precipitation seasonality are not direct results of our analysis, but rather stem from the input climate scenarios. This is now discussed in more detail in section 2.2.2. Additionally, the variation of climate change impacts on extreme head across different types of canal dikes is discussed in 4.3.2, where the results are explained, as also discussed in previous responses. Furthermore, this is discussed in the second paragraph of the Discussion section 5.2, where these findings are compared to other literature. We have added some finalizing sentences in this paragraph of the manuscript to highlight the importance of different head responses of canal dikes when considering future climate conditions.